# Insights into wind turbine reflectivity and RCS and their variability using X-band weather radar observations

Martin Lainer[1], Jordi Figueras i Ventura[1], Zaira Schauwecker[1], Marco Gabella[1], Montserrat F.-Bolaños[2], Reto Pauli[3], and Jacopo Grazioli[1]

[1]Federal Office of Meteorology and Climatology, MeteoSwiss, Locarno-Monti, Switzerland
[2]Federal Office for Defence Procurement, armasuisse, Science and Technology, Sensorik, Switzerland
[3]Military Aviation Authority (MAA), Switzerland

**Correspondence:** Martin Lainer (martin.lainer@meteoswiss.ch)

**Abstract.** The increasing need of renewable energy fosters the expansion of wind turbine sites for power production throughout Europe with manifold effects, both on the positive and negative side. The latter concerns, among others, radar observations in the proximity of wind turbine (WT) sites. With the aim of better understanding the effects of large, moving scatterers like wind turbines on radar returns, MeteoSwiss performed two dedicated measurement campaigns with a mobile X-band Doppler polarimetric weather radar (METEOR 50DX) in the north-eastern part of Switzerland in March 2019 and March 2020. Based on the usage of a X-band radar system, the performed campaigns are up to now unique. The main goal was to quantify the effects of wind turbines on the observed radar moments, to retrieve the radar cross section (RCS) of the turbine themselves, and to investigate the conditions leading to the occurrence of the largest RCS. Dedicated scan strategies, consisting of PPI (Plan Position Indicator), RHI (Range-height Indicator) and fixed-pointing modes, were defined and used for observing a wind park consisting of three large wind turbines. During both campaigns, measurements were taken in $24/7$ operation. The highest measured maxima of horizontal reflectivity ($Z_H$) and RCS reached $78.5\,\mathrm{dBZ}$ respectively $44.1\,\mathrm{dBsm}$. A wind turbine orientation (yawing) stratified statistical analysis shows no clear correlation with the received maximum returns. However, the median values and $99^{th}$ percentiles of $Z_H$ show different enhancements for specific relative orientations. Some of them remain still for Doppler filtered data, supporting the importance of the moving parts of the wind turbine for the radar returns. Further, we show, based on investigating correlations and an OLS (ordinary least square) model analyses, that the fast changing rotor blade angle (pitch) is a key parameter, which strongly contributes to the variability of the observed returns.

## 1   Introduction

The rapid development and expansion of wind farms in the latest years are a significant source of concern for the weather (e.g., Norin, 2017) and aviation radar community (de la Vega et al., 2016; Cuadra et al., 2019). Wind turbines are very large, reflective and moving objects, which makes them a source of clutter that becomes difficult to filter or separate from return signals of interest. Over the last years, the demand for the quantification, modeling and mitigation of the effects of wind turbines on radar systems is rising as the number of installed, planned or foreseen wind turbines is highly increasing. As analyzed by Komusanac et al. (2020) in 2019 about $15.4\,\mathrm{GW}$ of new wind power capacity was installed in the European Union (EU).

This is 27 % higher than in 2018. The total capacity of wind energy in the EU in the end of 2019 is 205 GW. The effective real production amounts to about 417 TWh, which is 15 % of the total consumed electricity. With the fact that green energy is becoming trendy, a realistic outlook until 2030 is to have around 323 GW of wind turbine power installed (Nghiem et al., 2017). By assuming a perfectly rational market here, it is expected that more wind turbines pop up.

Several studies exist in the literature about the evaluation and quantification of the impact of wind turbines on radar systems. These studies discussed the issue of clutter contamination of weather radar data (Lepetit et al., 2019; Hood et al., 2010; Angulo et al., 2015) as well as the identification of adverse effects of wind turbines on the performance of air surveillance and marine radars (Angulo et al., 2014; Cuadra et al., 2019). In general, wind turbine clutter reflectivity is depending on various parameters such as wind turbine dimensions, incidence angle of radiation, rotor speed, nacelle orientation and radiation frequency (Gallardo-Hernando et al., 2011; Norin, 2015).

A key parameter for the evaluation of how efficiently electromagnetic waves interfere with a physical object is the radar cross section (RCS). RCS is an optimal variable to estimate the effect of a "point target" on the performance of a radar system. With the term "point" we mean a target, which is much smaller than the radar sampling volume and with a size such that the incident field could be assumed to be planar over the whole extent of the target. However, the RCS concept is often generalized and extended to larger objects, starting with small airplanes, but then reaching even large airplanes and wind turbines. As a matter of fact, existing numerical models for estimating the back-scattering efficiency of wind turbines rely on this quantity. It is the projected area needed to isotropically re-irradiate the same power as the target scatters in the direction of the receiver and is usually expressed in decibel units related to one square meter (dBsm) (Knott et al., 2004; Skolnik, 1990). The detailed background on how the RCS is computed within our system is given in Sec. 3. A lot of studies have been published evaluating the RCS of individual wind turbines and wind farms and the effects on radar and communication systems. For instance, Lute and Wieserman (2011); Kong et al. (2011); Kent et al. (2008) have used measurements to characterize wind turbine scattering properties and the impact on radar performance. Others used numerical tools to investigate RCS and Doppler signatures of model-based wind turbines (de la Vega et al., 2016; Muñoz-Ferreras et al., 2016; He et al., 2015). The electromagnetic interactions between wind turbines and radar signals are complex and the general understanding still limited.

In this work the effort was put on the analysis of the data of two dedicated field campaigns which took place in March 2019 and March 2020, aiming at gathering weather radar measurements in the X-band frequency of three large wind turbines. The primary focus of this paper is to present a statistical analysis of radar reflectivity ($Z_H$) and retrieved RCS values and to find the relation between those variables and the operational data of the wind turbines (orientation, blade pitch angle, revolution speed).

In Section 2 we describe the field campaigns and X-band weather radar system used in this study (METEOR 50DX) as well as some key parameters of the wind turbine targets. More details on the observation site and visibility towards the wind park are given in Sec. 2.1. The special scanning strategies are the topic of Sec. 2.2, while the data sets are briefly treated in Sec. 2.3. In Section 3 we present, first, global statistics, regarding unfiltered horizontal reflectivity $Z_H$ for all three wind turbines during the field campaign in 2019. For simplicity, we call $Z_H$ just horizontal reflectivity hereinafter for which no Doppler or any other quality filter is applied. With data from the March 2020 campaign, the impact of the relative (with respect to the radar location) nacelle orientation (yaw angle), blade orientation (pitch angle), rotor revolutions and wind speed on the received

**Table 1.** Specifications of the wind turbines in the observed wind park during the measurement campaigns in 2019 and 2020.

| Type | Nordex SE N131/3300 |
| --- | --- |
| Rotor diameter | $131\,\mathrm{m}$ |
| Total height | $199.5\,\mathrm{m}$ |
| Hub height | $134\,\mathrm{m}$ |
| Rated Power | $3300\,\mathrm{kW}$ |
| Cut-in wind speed | $3\,\mathrm{m\,s^{-1}}$ |
| Cut-out wind speed | $20\,\mathrm{m\,s^{-1}}$ |
| Rated rotational speed | $10.9\,\mathrm{rpm}$ |
| Tip speed (max) | $74.8\,\mathrm{m\,s^{-1}}$ |

radar returns is investigated by different correlation analyses (Sec. 3.2). The impact of a Doppler notch clutter filter on parts of
the post-processed data is discussed. To even better explain the variance of the maximum $Z_H$, an OLS (ordinary least square)
model fit is performed (Sec. 3.3). Finally, in Sec. 4 a summary and conclusion are provided.

## 2 Methods and data

The main goal of the measurement campaigns in 2019 and 2020 was to study the interaction between electromagnetic waves
at $9.84\,\mathrm{GHz}$ sent by the weather radar and the structure of wind turbines. For this type of investigation, a small wind park
consisting of three large wind turbines, with the characteristics specified in Table 1, was selected near the city of Schaffhausen.
Many logistical requirements for the installation of the weather radar had to be met (e.g. sufficient power supply, radiation
safety, permissions from the Swiss Federal Office of Communication), beside a good visibility towards the wind park. More
details on the observation site are provided in Section 2.1.

The field campaign in 2019 took place between $6^{th}$ and $28^{th}$ of March with totally 23 days of continuous observations,
while the second one took place between $4^{th}$ and $25^{th}$ of March 2020 with a total of 22 days of measurements. During the
latter campaign, the radar data were collected with a fixed-pointing antenna towards the nacelle of the closest wind turbine
with respect to the radar site. The radar scanning protocol lasted $120\,\mathrm{min}$, whereof $100\,\mathrm{min}$ were used for the fixed-pointing
measurements and about $12\,\mathrm{min}$ for a PPI volume acquisition as an overview scan for the whole wind park area.

The measurements presented in this paper have been collected with a dual-polarization, simultaneous transmission and
reception (STAR), mobile Doppler weather radar, which operates at a frequency of $9.48\,\mathrm{GHz}$ (X-band). This radar system is
sensitive mostly to hydrometeors in the precipitation-size range. Due to the relatively small antenna size and overall weight,
it is a transportable system integrated on a trailer and particularly suitable for field campaigns and agile relocations. Several
configurations of the transmission protocols and data acquisition can be defined (e.g. PRF, pulse widths, scan velocity, data
acquisition rate). For the campaigns we are consistent and stick to one pulse width of $0.5\,\mathrm{\mu s}$ to have a good target range

**Table 2.** Specifications of the mobile METEOR 50DX dual-polarization weather radar used for the measurement campaigns in 2019 and 2020.

| Parameter | Specification |
| --- | --- |
| Frequency | $9.48\,\mathrm{GHz}$ |
| Transmitter type | Coaxial magnetron |
| Transmit power per H/V channel | $\sim 36\,\mathrm{kW}$ |
| Receiver linearity | $90\,\mathrm{dB} \pm 0.5\,\mathrm{dB}$ |
| Half power beam width $(3\,\mathrm{dB})$ | H: $1.25°$ |
| Gain (minimum) | $> 42.5\,\mathrm{dB}$ |
| Radome | Yes (screwed) |
| Elevation scan range | $-1°$ to $181°$ |
| Azimuth scan range | $0°$ to $360°$ |
| Pointing accuracy | $0.1°$ |

resolution of $75\,\mathrm{m}$, where most of the radiation energy is scattered by the WT object. The $0.5\,\mu\mathrm{s}$ pulse shape is, compared to the one for $0.33\,\mu\mathrm{s}$, more uniform and thus preferred in our system. The antenna movement for the measurements in 2019 is slow, ensuring data collection every $0.1°$ in azimuth (one radial), while the PRF is set high enough $(2\,\mathrm{kHz})$ to ensure a large number of pulses for each radial and a reasonably good unambiguous velocity range. A detailed technical overview about the characteristics of the radar system can be found in Neely III et al. (2018). Some key specifications are listed in Table 2.

The radar system provides a set of single-polarization, dual polarization, and Doppler measurements: horizontal (vertical) reflectivity $Z_H$ ($Z_V$), differential reflectivity $Z_{DR}$, co-polar correlation coefficient $\rho_{HV}$, total differential phase shift $\Phi_{DP}$, specific differential phase shift $K_{DP}$, Doppler velocity $V$ and Doppler spectrum width $W$. If filters or thresholds are applied to the data, not only the filtered data can be kept, but always also the raw data. Additionally it is possible to store full power spectrum (PSR) data.

After the wind turbine campaign in 2019 two main upgrades of our mobile radar system have been conducted. In June a complete new seamless radome has been installed. The improvements compared to the former radome, which was joint by metal units, are shown and discussed in Figueras i Ventura et al. (2020b). Concerning power-related measurements, it is important to know that the attenuation of both radomes have similar values. Later in October, an important software upgrade was performed, allowing now to acquire data when the radar antenna is not moving (something that was not possible to do in 95 2019). This new ability is hereafter referred to as fixed-pointing or stare-mode measurements of the weather radar.

The core data processing for both campaigns was done by the MeteoSwiss in-house-developed open-source real-time weather radar data processing framework Pyrad (Figueras i Ventura et al., 2020a), which is based on the Py-ART radar toolkit (Helmus and Collis, 2016).

## 2.1 Observation site and radar visibility

The observation site, where the radar system could be installed, was in the vicinity of the city of Schaffhausen. This site had several advantages: line of sight with the wind turbines, a minimum distance from the wind park to limit the risk of a receiver saturation, site accessibility, permission to transmit and power connection. The three wind turbines of the small wind park located north of Schaffhausen are installed on a hill surrounded by forests. The corresponding terrain profile between the radar and the furthermost wind turbine (WT3) is shown in Fig. 1c. The locations of the turbines as seen from the radar site are at

distances of $7.7\,$km, $8.1\,$km and $8.6\,$km with directions of $337.8°$, $343.3°$ and $340.2°$ from North. In order to hit the turbine ground locations, elevation angles of $2.24°$ (WT1), $2.1°$ (WT2, WT3) are needed.

Although trees were blocking part of the radiation towards the masts at a distance of $1\,$km, the rotor centers of all three wind turbines were always visible at the center of the radar antenna beam. With radar ground echo clutter simulations based on GECSX (Ground Echo Clutter Simulator) and a digital elevation model (DEM) with $50\,$m resolution, the radar visibility

towards the wind turbines could be determined. The used approach follows the technique and developed software described in Gabella and Perona (1998); Gabella et al. (2008). Figure 1a shows the minimal visible elevation map, which is the minimum radar antenna beam elevation angle to get maximum $50\,\%$ beam blockage, starting from the weather radar observation site. The locations of the wind turbines are indicated as three white plus signs in the maps, while the radar location is shown as a red plus sign. Out of the first map (Fig. 1a) we get the lowest elevation angles with visibility at the center of the radar beam: $2.25°$

(WT1), $2.10°$ (WT2) and $2.15°$ (WT3). With increasing elevation of the radar beam the visibility gets better (s. Fig. 1b). At an elevation of $3°$ all three turbine locations are visible with the whole HPBW (half power beam width) angle.

Given the distances to the wind turbines, ranging between $7.7$ to $8.6\,$km with respect to the radar site, together with the characteristics of the radar antenna (HPBW of $1.25°$), a beam broadening between $175$ and $195\,$m can be assumed at the ranges of the wind turbines. Compared to the spatial dimensions of the wind turbines, the beam shape is already larger when it is

120 reaching the target. Due to the fact that the radar measurements are the result of a convolution between the beam shape and wind turbine structures, it is difficult to distinguish return signals from structures that are smaller than the beam shape during the radar scans.

## 2.2 Radar scanning strategy

The scanning strategy of the weather radar system in 2019 consisted in the continuous repetition of a protocol lasting $45$

minutes, including three different scan types (PPI, RHI and solar scan). While the PPI and RHI scans were dedicated to the wind turbine measurements, the solar scan is used for the receiver quality control. One PPI sector scan sequence lasted $12\,$min, one RHI scan sequence took $18\,$min.

The sequence of PPI scans is a narrow volume scan in the azimuth sector $331°$ to $349°$ consisting of 30 individual PPIs at increasing elevation angles. Regarding the RHI scans a sequence of $81$ individual RHIs were set up. The RHI scans are

130 separated by $0.2°$ at the edges of the azimuthal scan area and $0.1°$ in the core region. In order to optimize the mechanical antenna movements of the scans, the RHIs are conducted from the smallest to the largest azimuth angle with alternating

**Table 3.** Scanning parameters of the mobile weather radar used for the wind turbine measurement campaign in 2019. The solar scan parameters are not included.

| Parameter | PPI | RHI |
|---|---|---|
| Azimuth range (resolution) | 331–349° (0.1°) | 335–336.6° (0.2°), 336.7–342.9° (0.1°), 343–344.6° (0.2°) |
| Elevation range (resolution) | 2.0–4.0° (0.1°), 4.1–5.1° (0.2°) | 2–10° |
| Range (resolution) | 0–30 km (75 m) | 0–30 km (75 m) |
| Antenna speed | $1° \, \mathrm{s}^{-1}$ | $1° \, \mathrm{s}^{-1}$ |
| PRF | 2 kHz | 2 kHz |
| Pulse width | 0.5 μs | 0.5 μs |
| Nyquist velocity | $15.82 \, \mathrm{m \, s}^{-1}$ | $15.82 \, \mathrm{m \, s}^{-1}$ |
| Scan duration | 12 min | 18 min |

upward and downward antenna motions which create a continuous sampling of the volume. With the use of RHI scans a better characterization of the influence of the secondary lobes towards the ground is possible.

The detailed parameters of both scan types are listed in Table 3. The movement of the antenna is slow for all scans, ensuring data collection every 0.1° (one radial) in azimuth, while the PRF (Pulse Repetition Frequency) is high enough to ensure a large number of pulses for each radial and a reasonably good unambiguous (Nyquist) velocity range. No speckle or spatial filter were applied with the data acquisition. A Zero-Doppler filter has only been used to post-process some of the gathered power spectra during PPI scanning modes.

With the upgraded system a new scanning method, the fixed-pointing, could be applied in 2020, which should complement the results obtained a year before, by help of a very high temporal resolution of the acquired data in stare-mode. During the fixed-pointing scans, the antenna of the radar was not moving and always pointing to the same wind turbine. For the whole campaign in 2020, we observed only that wind turbine (WT1), where the visibility was best. One goal we wanted to achieve with the stare-mode was to have a complete overview of the variations in time and revolution cycles of the blades. This was not possible to achieve with the slow scanning strategy in PPI and RHI modes of 2019. However, every 2 hours a PPI volume scan similar to the ones during the first campaign was taken in order to get a broader overview across the wind farm area. The data acquisition time for the fixed-pointing measurements is as short as 64 ms. Other relevant characteristics of this new radar scan mode are summarized in Table 4.

## 2.3 Weather radar and wind turbine data

Over the entire measurement campaign in 2019 we gathered in total 612 full PPI and 700 full RHI volume sector scans covering the wind park during the 23 measurement days with its various environmental conditions. The month of March 2019 had a reasonable variability in terms of weather. Wind conditions have been particularly interesting also due to many periods of strong winds. The wind rose plots in Fig. 2 show how the wind speeds and directions were stratified in March 2019 (a) and

**Table 4.** Scan parameters of the fixed-pointing mode of the weather radar in 2020.

| | |
|---|---|
| Azimuth angle | 338.9° |
| Elevation angle | 3.1° |
| Range (resolution) | 0–15 km (75 m) |
| Antenna speed | $0°\,s^{-1}$ |
| PRF | 2 kHz |
| Pulses per ray | 128 |
| Aquisition time | 64 ms |
| Pulse width | 0.5 μs |
| Nyquist velocity | $15.82\,m\,s^{-1}$ |
| Scan duration | 100 min |

2020 (b) as seen from a nearby wind profiler. Two main direction modes (from north-east and south-west) are dominating the statistics and this is also visible in the temporal evolution of the relative position (with respect to the radar location) of the turbine nacelles (Figs. 3b and 3c). In this Figure we representatively show the data for wind turbine WT1. As seen relative to the radar angle of attack, the main orientation is centered between 50° and 100°. The rotor speeds $r_s$, which are presented in revolutions per minute, are plotted on top in red and reveal that a $r_s$ of about 11 rpm is the main power production operation mode of this kind of turbine. The data from the wind turbines has a temporal resolution of 10 minutes and was kindly provided by the wind park operator.

In order to illustrate the narrow sector PPI scans of the weather radar in 2019, the four plots in Fig. 4a-d are shown. At an elevation of 3° PPIs of the horizontal reflectivity $Z_H$ (a), the differential reflectivity $Z_{DR}$ (b), the co-polar correlation coefficient $\rho_{HV}$ (c) and the Doppler spectrum width $W$ (d) are represented on 2019-03-24. The wind turbine clutter is made visible by the the black 35 dBZ reflectivity contours in all shown PPI moment data at the range from 7.7 to 8.6 km. The impact of turbine clutter cannot be seen in all PPIs clearly. While $Z_H$ shows values up to 40–50 dBZ for the given elevation of 3°, $Z_{DR}$ attains mostly slight negative numbers within the 35 dBZ contours. In those areas and in the shadow behind, $\rho_{HV}$ is reduced to approximately 0.7 compared to the hill forest clutter in front of the wind turbines, where $\rho_{HV}$ reaches more than 0.95. For this specific example and the slow PPI scan, the Doppler spectrum width remains close to $0\,m\,s^{-1}$ on average as indicated by the violet areas.

For the statistical analyses azimuth elevation plots at a fixed range or range span are used. Figures 5a-d show for a specific PPI (a) and RHI (c) scan sequence the maximum horizontal reflectivity within the range span of all the three wind turbines. Below 2° in elevation a software-based blanking was applied to the radiation transmitter for safety reasons. All three wind turbines can be distinguished in these kind of plots. WT1 (left signatures at 339°) with the best visibility has in this example the highest returns of about 60 dBZ within the center region where the nacelle is located. The benefit of the RHI-based results is the larger overview for elevations up to 10°. Figures 5b and 5d show the range gates from both scan sequences respectively, where the maximum of $Z_H$ occurred. With the distance of these range gates, it is obvious that the high returns left of WT1

are not associated to a turbine effect as they appear at a shorter distance from the radar of about $7500\,\mathrm{m}$ compared to the WT1 distance of $7740\,\mathrm{m}$. Given the surroundings consisting of uphill terrain (s. Fig. 1), the high radar returns have most likely the origin in background clutter from the hill forests.

In March 2020 the radar was placed exactly at the same location as the year before. The whole campaign lasted 22 days and was dedicated solely to the observation of wind turbine WT1. The radar beam center was targeted at the elevation of the nacelle and thus towards the center of the rotors. The detailed characteristics are summarized in Table 4. In total we gathered $2.429 \times 10^7$ data samples, where one sample (or one radial) consists of 128 averaged radar pulses. The full $Z_H$ data set is illustrated also later in Section 3.1 with Fig. 8. Together with the available environmental wind turbine data, which is available at a $10\,\mathrm{min}$ resolution scale, we aim at finding relations for the occurrence of radar return maxima and minima as well as their variability.

## 3    Statistical analysis of the weather radar observations

In the following part of the paper, all the horizontal reflectivity $Z_H$ measurements obtained from PPI and RHI scans in 2019 are statistically analyzed in order to characterize the returns from the three wind turbines by using the radar data processing framework Pyrad (Figueras i Ventura et al., 2020a). Global statistics considering the median and maximum $Z_H$ are presented. Given the stability of the location of the wind turbine signals, statistics can be computed at fixed ranges, on a two-dimensional plane that is given by azimuth (x-axis) and elevation (y-axis) (s. Sec. 3.1). While radar reflectivity values may be informative, a core information allowing to generalize the measurements collected is the retrieved RCS, usually given in $\mathrm{dBsm}$ (dB square meters). Its added value with respect to reflectivity alone is that it is a pure property of the target. The RCS values that will be shown here are retrieved from the inversion of the formula for the radar reflectivity $Z$. According to Battan (1973) the radar equation for a single, isolated target is:

$$p_r = \left( p_t G_0^2 \frac{\lambda^2}{(4\pi)^3} \right) \frac{\sigma_b}{r^4} \tag{1}$$

where $[p_r] = \mathrm{mW}$ is the received lossless power by a directional antenna.

The log-transformed power can be derived by $P_r = 10 \cdot log(p_r/p_0)$, where $p_0 = 1\,\mathrm{mW}$. In Equation 1, the backscattering cross section or RCS is indicated as $\sigma_b$ (assuming a monostatic radar with a scattering angle of $-180°$ for the angular RCS), the antenna gain as $G_0 \approx \frac{\pi^2}{\theta_H \theta_V}$, the radar wavelength as $\lambda$, and the transmitted power as $p_t$. $\theta_H$ and $\theta_V$ are the horizontal and vertical beam widths in radians. Usually $\sigma_b$ is expressed as a scalar and the dependencies on wavelength is implicitly assumed: in our case $\lambda = 0.032\,\mathrm{m}$. $[\sigma_b] = \mathrm{m}^2$, respectively RCS $= 10 \cdot log(\sigma_b/\sigma_0)$, where $\sigma_0 = 1\,\mathrm{m}^2$.

Precipitation particles act as distributed scatterers in the volume of air illuminated by the weather radar. The resolution volume $V_r$ filled by a transmitted pulse can be approximated by a cylindrical shape, given the radar pulse width $\tau$ and the range $r$:

$$V_r \approx \pi \left( r \frac{\theta_H}{2} \right) \left( r \frac{\theta_V}{2} \right) \frac{c\tau}{2} \tag{2}$$

Accounting for the actual distribution of power within the beam generated by a parabolic antenna, a correction factor of $1/(2 \cdot ln\,2)$ was introduced by Probert-Jones (1962):

$$V_r = \frac{\pi r^2 c\tau \theta_H \theta_V}{16 \cdot ln\,2} \tag{3}$$

The backscattered signal from a volume of randomly distributed scatterers is the sum over all scattered signals. The summation of the backscattered cross sections from precipitation scatterers in a unit volume is called the radar reflectivity and is defined as $z = \sum_i \sigma_{b,i} n_i$. The final form of the radar equation for distributed scatterers, like precipitation, combines Equations 1 and 3 and substitutes $z$ in the resolution volume, $V_r \sum_i \sigma_{b,i} n_i$, for $\sigma_{b,i}$. The general form of the radar equation that is valid for scatterers of all sizes yields:

$$p_r = \frac{p_t G_0^2 \lambda^2 \theta_H \theta_V c\tau}{1024(ln\,2)\pi^2 r^2} \sum_i \sigma_{b,i} n_i \tag{4}$$

The general assumption behind the formulation presented in Equation 4 is that of a pulse volume homogeneously filled by a huge number of randomly distributed backscatterers. For a weather radar the average returned power has to be related to the physical characteristics of the particles within the resolution volume. The Rayleigh approximation of the backscattering cross section of a single water drop $\sigma_{b,i}$ can be expressed, according to e. g. Fabry (2015) or Ryzhkov and Zrnic (2019), as

$$\sigma_{b,i} = \frac{\pi^5}{\lambda^4} |K|^2 D^6, \tag{5}$$

where $|K|$ is related to the complex index of refraction. By substituting Equation 5 into Equation 4 the radar equation for spherical drops can be written as:

$$p_r = \frac{p_t G^2 \lambda^2 \theta_H \theta_V c\tau}{512(2\,ln\,2)\pi^2 r^2} \cdot \frac{\pi^5 |K|^2}{\lambda^4} \sum_i D_i^6 n_i \tag{6}$$

The conversion to the wind turbine radar cross section $\sigma_b$ that has been used in our study is finally shown by Equation 7.

$$\sigma_b = \frac{\pi^6 c\tau \theta_H \theta_V}{16\,ln\,2} \frac{|K|^2}{\lambda^4} r^2 z \tag{7}$$

**Table 5.** Conversion factors $F$ to obtain the radar cross section RCS from $Z_H$ for the three wind turbines WT1, WT2 and WT3.

| Wind turbine | $F$ |
|:---:|:---:|
| WT1 | $34.41\,\mathrm{dB\,mm^6\,m^{-5}}$ |
| WT2 | $34.08\,\mathrm{dB\,mm^6\,m^{-5}}$ |
| WT3 | $33.50\,\mathrm{dB\,mm^6\,m^{-5}}$ |

The three conversion values $F$, which have to be subtracted from the retrieved $[Z] = \mathrm{dBZ}$ to obtain $[\mathrm{RCS}] = \mathrm{dBsm}$ for all wind turbines, are summarized in Table 5. The reflectivity in linear unit $[z] = \mathrm{mm^6\,m^{-3}}$ is given by $z = \sum_i D_i n_i$ (Bringi and Chandrasekar, 2001), where $[n_i] = \mathrm{m^{-3}}$.

With the much better temporal resolved measurements acquired in 2020 with the fixed-pointing scan mode of the weather radar, the impact of the turbine orientation, blade pitch angle $\theta$, rotor speed $r_s$ and wind speed $\bar{U}$, as a measure of blade bending, on the retrieved horizontal reflectivities is investigated later in Sec. 3.2. In a first step the relative turbine (nacelle) orientation is solely used to stratify the returns with an azimuth bin of $10°$ width. Following, in order to measure the strength and direction of possible relationships between the different WT data sets, we perform a correlation analysis based on the Pearson correlation coefficient. In order to simplify the use of a linear regression, we converted the relative nacelle positions $\alpha$ to normalized positions $\Psi$, such that values between 0 and 1 map symmetrically independent between backward or forward and sideways facing of the WT nacelle. We calculate $\Psi$ simply as $\Psi = |\sin\alpha|$. For clarification, a $\Psi$ value of 0 means that the turbine's angle of attack is either facing towards the radar or away from the radar while a value of 1 shows the sideways case when the angle of attack is perpendicular to the radar beam. With the $\Psi$ parameter a more reasonable calculation of the linear Pearson correlation coefficient is achievable because the polar $360°$ projection of the turbine position alone has a symmetry.

## 3.1 Global statistics of horizontal reflectivity $Z_H$ and radar cross section RCS

The global statistics are computed at a range distance of $7740\,\mathrm{m}$ (WT1), $8040\,\mathrm{m}$ (WT2), and $8600\,\mathrm{m}$ (WT3) over the entire data set of the 2019 field campaign. The statistical overviews in Figs. 6 and 7 show the maximum (1st row) and median (2nd row) of $Z_H$ for all three wind turbines. The used data set to compute the statistics for Fig. 6 refers to all PPI scans, while for Fig. 7 all RHI scans were processed. As it can be seen in those images, the visibility over the wind turbines was very good for the selected site. At elevation angles lower than $2°$ sector blanking has been applied, where the radar transmission was turned off for safety reasons. While the elevation span of the PPI scans is enough to fully observe the wind turbines, the increased vertical span of the RHI scans allowed the observation of the returns that are not associated to the main lobes of the radar antenna.

A clear example can be seen in the median value of $Z_H$ within Fig. 6. Secondary returns appear in the area surrounding the core of the wind turbines, either generating the visual effect of a ring-signature around the most intense echoes, or as secondary/tertiary replications of the intense signals. Those returns are 20 to $50\,\mathrm{dB}$ lower than the strongest returns from the turbines themselves. These signals remain significant and populate a large sector of azimuth and elevation angles around the actual location of the wind turbines, thus extending the area which has to be seen as WT clutter-contaminated. The character-

istics of secondary or tertiary returns are obviously depending on the characteristics of the antenna (shape, secondary lobes, symmetric/an-isotropic behavior) of each individual radar.

Based on Fig. 6, we find the largest area of maximum returns ($Z_H$) spreading from the rotor center of wind turbine WT1, which is likely related with the visibility of the radar beam. Regarding $Z_H$, WT1 reached also the maximum of about $75\,\text{dBZ}$ in the RHI data during the first campaign in 2019. By looking to the fixed range contour plots of WT1, a very high and stable return signal can be identified to the lower right at about $341°$ in azimuth and $2.5°$ in elevation direction. So far this signal could not be assigned to a source. Multi-path effects could be a possibility. By looking at the range gates in Fig. 4d it is evident that this peculiar signal has its maximum at the same range as the wind turbine WT1, and is thus likely related to the interference with this turbine.

While the location of the intense signals is stable in time, the intensity of the returns varies significantly. This is evident when we look at the differences between the maximum and median $Z_H$ values, which are on the order of $20\,\text{dB}$. One of the main objectives of the measurement campaign is the evaluation of the most critical scenarios in terms of high wind turbine radar cross sections. Here we summarize the highest observed values and their frequency of occurrence. The highest RCS values retrieved have been on the order of $40\,\text{dBsm}$ for the RHI-based data set and $38\,\text{dBsm}$ for the PPI-based one. These high values were very rare to observe, only a few echo counts could be assigned to them. WT1, with the best radar visibility, reached the highest values. The variability of the intensity of the maximum returns was found to be significant, and the maximum RCS of each radar scan can be as low as $20\,\text{dBsm}$ (or lower). The underlying dynamics is given by the combination of the changing turbine orientation, motion of the rotors and blade pitch angles together with the motion of the radar antenna, which scans over the wind park and thus is only covering the same exact spot at intervals of several minutes. This was the main motivation to introduce fixed-pointing observations during the second field campaign in 2020. The amount of data produced with the fixed-pointing acquisition mode was as high as one radial collected every $64\,\text{ms}$. This increases the confidence that actual maxima of the radar returns are captured within our long dedicated observation period towards WT1. In order to give an idea about the amount of data that was captured the scatter plot in Fig. 8a is visualized, showing the whole time series of $Z_H$ (in black) and the corresponding $10\,\text{min}$ moving median values (in red).

The variability of the returns remains extremely high, with typical short-term spans on the order of $20\,\text{dB}$. A first conclusion is that the variability which has been observed during the 2019 field campaign was not caused by the scanning strategy itself. With the radar stare-mode, frequently higher maxima than during the PPI and RHI scans of 2019 were observed in the reflectivity data. Thus the usage of fixed-pointing data acquisition indeed improved the sampling of the right tail of the distribution of $Z_H$ values. This is especially important, when worst case scenarios are of interest, e.g. for aviation safety issues. $Z_H$ maxima of $78.4\,\text{dBZ}$ (equivalent to an RCS of $44\,\text{dBsm}$), about $4\,\text{dB}$ higher than what has been observed in 2019, were measured. From the time series shown in Fig. 8 some interesting periods can be identified, when the spread between the maxima and minima of $Z_H$ is suddenly increasing. Between 2020-03-14 and 2020-03-21 such sudden increases were more frequently observed. In Sec. 3.2 we investigate these findings in depth by performing correlation analyses with i. a. the pitch angle $\theta$ of the rotor blades. The $10\,\text{min}$ moving median mostly stays between 50 and $60\,\text{dB}$ over the 2020 campaign period, occasionally going up to 65–70 dB.

Having access to spectral data, it is worth to investigate the potential of frequency-domain filters to suppress the returns of the wind turbines. Thus, we tested a spectral clutter suppression of width $7\,\mathrm{m\,s^{-1}}$ (suppressing the returns in the Doppler interval $\pm 3.5\,\mathrm{m\,s^{-1}}$). The spectral clutter suppression shown here is simple: the contribution of power corresponding to the Doppler velocities within the width is removed. Data collected in 2020 allow for a finer comparison of filtered (clutter-suppressed) $Z_{Hf}$ data and unfiltered ones. First, it can be observed how the operational characteristics of WT1 affect the efficiency of clutter removal. And second, the corresponding removal of static parts like the WT tower and mostly the nacelle allows to assign the seen effects in the radar returns to the remaining moving rotor blades.

Doppler spectra have been collected every 5 minutes for 5 minutes during the field campaign 2020, with a few days of data gaps given by the very high demand in terms of data computation and transfer on the signal processor. However, we can get a clear idea of how a clutter-notch based filter was performing during the entire campaign (s. Fig. 8b). The clutter suppression is very efficient as soon as the rotor speeds get close to $0\,\mathrm{rpm}$, and most of the signal can be removed. The median residual reflectivity is on the order of $20\,\mathrm{dBZ}$ in those cases. The 0-Doppler component of the wind turbines is very high even when the turbine is spinning. In this way, the clutter notch filter can remove a part of the signal even during normal operation of WT1. Looking at the plots, this reduction is on the order of $10\,\mathrm{dB}$ in median terms. Overall a significant attenuation, but one must keep in mind that the residual signal is still very high.

## 3.2 Impact of WT orientation, blade pitch angle, rotor speed and wind speed on $Z_H$

Given the availability of the WT and wind sensor data it is possible to perform turbine orientation stratified statistics with $Z_H$ and $Z_{Hf}$. The first experiment, which has been conducted in 2019, provided non-conclusive results with respect to the orientation of the wind turbines and this was another reason that motivated a second experiment in 2020. Now, with the high temporal resolution data, we are more confident about the obtained results that we intend to show here. Both, March 2019 and March 2020 had similar statistical meteorological wind patterns (s. Fig. 2), which led to comparable distributions of the relative nacelle orientations (s. Figs. 3b and 3c). A different view on the relative positions are provided in Figure 9, where the normalized polar distribution are shown in combination with the rotor speeds for the $Z_H$, respectively $Z_{Hf}$ data. To clarify a bit the relative nacelle positions, a value of $0°$ means that the long axis of the rotor blades is perpendicular aligned to the direction of the radar beam center and the WT is pointing towards the radar (left WT schematic in Fig. 3a). On the contrary, a value of $180°$ represents a WT pointing away from the radar and with $90/270°$ the long axis of the blades is parallel aligned to the center of the radar beam (right WT schematic in Fig. 3a).

From the polar plots in Fig. 10a we see that the maximum measured reflectivities are insensitive to the relative yaw angle of the nacelle. This is evident by the round and not too much disturbed distributions (less than $10\,\mathrm{dB}$) of the color-mapped scatter points. In this plot the maximum $Z_H$ values are counted within a bin width of $0.5\,\mathrm{dB}$. Biased by the wind direction distribution in March 2020, most maxima were retrieved at $270°$ marked by the yellow scatter point, with a total number of 10 data values. By having a look to the $99^{th}$ percentile distributions together with the number of counts exceeding the plotted values of the scatter points (inner color-mapped scatter plot circle), first deformations of the distribution in the polar plots become visible and even more in the median distribution (red scatter points). It is interesting to observe how, in median terms

the relative orientations near $180°$ are associated to higher returns, while for $99\%$ quantiles, similar signals are observed also at orientations near $90°$ and $270°$.

With Fig. 10b ($Z_{Hf}$ data only) in comparison we try to separate the effects of the moving parts, which is mainly the rotor blades, from the static returns from the tower and nacelle. We assume that the returns from the slow moving nacelle are significantly reduced by the applied filter. In the polar distribution we find that the maximum $Z_{Hf}$ values get reduced by 5–$10\,\mathrm{dB}$ and reach mostly around $70\,\mathrm{dBZ}$, still rather independent of the relative WT1 orientation. The median distribution for $Z_{Hf}$ drops remarkably at two positions, $170°$ and $350°$. But this result is statistically not significant as the data samples for those directions are in general small (below $2\%$) and further decreased for the filtered reflectivity data (s. Fig. 9b). A higher reduction ($> 10\,\mathrm{dB}$) is found at $260°$ for the $99^{th}$ percentile when the turbine is pointing rightwards (wind direction from the east), but not for the opposite direction (WT pointing leftwards). One explanation can be that the radar beam center is filled with more moving parts of the turbine when the WT is pointing leftwards. This could be the case when the moving rotors are spinning in front of field of view towards the tower or nacelle.

Figure 11 compares on a 10 minutes moving window the maximum, minimum and median of filtered and unfiltered reflectivity. The separation of the individually colored scatter points indicates the two modes of the turbine, moving and not/slow moving. As naturally expected, the filter is more efficient when WT1 is not or slowly moving. The fact that the difference between $Z_{H,max}$ and $Z_{Hf,max}$ is smaller than the difference of the median or minimum is an additional sign that the high $Z_H$ come preferred from the moving WT parts. Regarding the possible influence of the WT nacelle, it can be argued that the effective scattering area of the WT nacelle is small compared to the area of the rotor. In the end this could lead to a stealth behavior, as long as the radar angle of attack is not perpendicular to the nacelle surface which is given for our measurement setup. Future work should take also into account numerical simulations to assess the complex interactions in more detail.

The complexity of an operating wind turbine is not only related to the nacelle, respectively rotor orientation (yaw angle) but also to the varying blade pitch angles $\theta$, which are used to control the rotor speed (power production) when the wind speed is changing. The profile of the blade rotates outgoing from the hub towards the blade tip in order to maintain the angle of attack (Gipe, 2004). $\theta$ is a parameter that may vary at the time scale of the seconds. Unfortunately, the granularity of the available wind turbine data ($10\,\mathrm{min}$ resolution) does not allow us to match the observed maxima with the actual wind turbine parameters at the exact same time, but while looking at how the maxima and minima appear in the data (s. Fig. 12a) it is reasonable to assume that the blade angle is a key parameter. From the time series plots themselves, a linkage between the blade angles and moving $Z_{H,max}$ (green line) and $Z_{H,min}$ (blue line) signatures is obviously present during certain time periods. For example on 2020-03-12 or 2020-03-22 blade pitch angles of WT1 were aligned at roughly $90°$ over hours and at the same time the maximum $Z_H$ values increased by more than $10\,\mathrm{dBZ}$ accompanied by a heavy increase in the spread between the minima and the maxima which is equivalent to an increase in the overall variability. It can be concluded that large changes in the blade angles over time, like in the time period between 2020-03-16 and 2020-03-21, lead to a strongly increased variability of the captured radar returns.

Another, unfortunately unsupervised, WT parameter which is supposed to influence the results is the bending or deflection of the blades, enhancing occasionally the radar returns. Realistic simulations by Zhang et al. (2018) assumed blade mid-deflection

angles of $15°$ for a $37\,\mathrm{m}$ blade length. Converted to larger turbines, like the Nordex SE N131, with blades as long as $65.5\,\mathrm{m}$ bending distances of $17.15\,\mathrm{m}$ at the blade tip level result if the deflection angle reaches $15°$. The signal enhancements are possible even if the wind turbine is not pointing towards the radar, as the blades are twisted and therefore mirror-effects are even possible sideways in azimuth, respectively elevation, from the main orientation of the rotational axis of the rotor. The blade bending effect is naturally closely related to the wind and rotor speed. The coupling between the bending and torsional deflection of a blade is generally known as aero-elastic tailoring (Weisshaar, 1981).

From the viewpoint of a MW-class turbine operator the stability of power production is important when delivering electricity into the grid. When the wind conditions are not good enough (e.g. too low wind speeds) and stable power production can not be assured anymore, the wind turbine is put into a so called sailing position. In such a position the pitch angle of the blades $\theta$ is highly increased ($\theta \sim 70°$) compared to normal operation, allowing the rotor to slowly turn (usually below $1\,\mathrm{rpm}$) while keeping the bearings of the system greased. Further the axial turbine load is constantly varying, which is also important to reduce the mechanical stress. During certain, mostly rare time periods, $\theta$ is put over $90°$ to aerodynamically break the rotors. Possible reasons are temporal fast changing gusts of wind, for instance during thunderstorm activity, or an external slow-down triggering to reduce i.e. bird and bat fatalities.

Next, a Pearson-based correlation analysis between the WT1 operation data and the radar returns is presented for the fixed-pointing measurements of 2020. One idea was to oppose the correlation results with data during a normal (power production) and abnormal (breaking or sailing) wind turbine operation. Therefore a threshold of $1\,\mathrm{rpm}$ for the rotor speed has been chosen to sub-divide the re-sampled ($10\,\mathrm{min}$) data sets of horizontal reflectivity ($Z_{H,max}$, $Z_{H,min}$, $Z_{H,r}$, $Z_{H,med}$) and WT parameters ($\theta$, $\Psi$, $r_s$, $\bar{U}$).For the moving maximum $Z_{H,max}$ (green line in Fig. 12a) and the pitch angle a weak positive Pearson coefficient $r$ of $0.32$ resulted. More interesting is the result between the moving minimum $Z_{H,min}$ (blue line in Fig. 12a) and $\theta$.

A theoretically strong downhill linear relationship with $r = -0.67$ can be calculated for them, but it has to be taken into account that the complex blade surfaces will likely not give a linear radar echo response which in turn would initiate some misleading result. In addition the abnormal turbine modes with less variable blade angles bias these results further. By only taking into account the data where $r_s \geq 1\,\mathrm{rpm}$ the correlation coefficient is reduced to $-0.56$. For $Z_{H,max}$ in contrast the change is small with $r$ reaching $0.34$. The corresponding correlation heat maps for both WT modes, separated by the defined rotor speed threshold of $1\,\mathrm{rpm}$, are presented in Fig. 13.

Furthermore, a strong uphill (positive) linear relations can be found between the rotor speed and the moving minimum $Z_{H,min}$ ($r = 0.63$) for the whole data and between the blade pitch angle $\theta$ and the min/max difference $Z_{H,r}$ ($r = 0.64$). From this point of view we deduce that $\theta$ is a good measure for the observed variability in the horizontal reflectivity of a wind turbine. Naturally, the rotor speed $r_s$ and $\theta$ have a very high anti-correlation ($r = -0.87$) and further it is interesting to see that without sub-dividing the data sets no significant relevant linear relation is evident between the $Z_H$ data sets (including the 99.9 and $0.1\,\%$ quantiles) and $\Psi$ (simplified relative nacelle orientation) or average wind speed $\bar{U}$ (measure of blade bending). By looking at the $r_s$ threshold split correlation heat maps in Figs. 13a and 13b, this picture changes. For the $r_s < 1\,\mathrm{rpm}$ case $\Psi$ now has a low positive correlation towards the maximum returns ($r = 0.33$) and the coefficients for $\bar{U}$ even reach $0.5$ (moderate uphill linear relation) when the turbine is in sailing or aero-breaking mode. On the other hand the $Z_{H,r}$ loses the high correlation

towards the pitch angle when the rotor is only slowly idling with the wind. This can be explained by the fact that $\theta$ is not much adjusting during these abnormal operation modes as it is evident in the scatter plot of Fig. 14a (vertical aligned scatter points for $r_s < 1\,\mathrm{rpm}$). In more detail the plots in Fig. 14 show 4 different color coded scatter plots between the blade pitch

angle $\theta$ or normalized relative yaw angle $\Psi$ and $Z_{H,max}$, respectively $Z_{H,min}$, below and above the rotor speed threshold of $1\,\mathrm{rpm}$. In addition, the linear regression lines with confidence interval sizes of $95\,\%$ for the regression estimate are drawn using translucent bands around the regression line. The maximum returns increase as the blade pitch $\theta$ increases, and this is especially true when the turbine rotors rotate slowly (sailing mode). We also see a high variability of $Z_{H,min}$ when $\theta$ is roughly at $70°$ or $90°$ ($r_s < 1rpm$). A positive correlation of $Z_{H,max}$ and negative correlation of $Z_{H,min}$ when the wind turbine is in normal

operation mode. The possibility that $Z_{H,max}$ is facing some radar saturation issue during the 2020 measurement campaign, it could be the case that the correlation coefficients are lower than they should be.

One question that arises is, what is causing the high variability of the minimum and maximum radar returns when $\theta$ is nearly constant (yellow and red scatter points in Fig. 14a). The remaining WT parameters are the nacelle orientation and possibly the blade bending related to the wind force. Both parameters are investigated in more detail with help of Figs. 14b and 15. By

405 investigating the $Z_H$ relations towards $\Psi$, the already presented positive Pearson correlation coefficient now is visible by the regression lines for the data sets with $r_s < 1rpm$ (comparable color codes as for $\theta$). It has to be added that a significant shift to stronger echoes is not described by the linear relation. Some part of the variability of the horizontal reflectivities during the abnormal slow rotating turbine modes can be ascribed to changes in the azimuth pointing of the turbine (yaw angle) which in turn changes from the radar location point of view the backscatter efficiency of the blades when the pitch is constant. The fact

that the radar observations were most of the time facing the WT from the side is clearly seen by the majority of scatter points beyond $\Psi$ values of $0.8$.

The second and more fuzzy and thus difficult WT parameter to assess is the blade bending. One way to increase the transformation of wind energy into power is to produce and use longer blades without a large increase in weight. Such blades, which are in use nowadays, tend to be less stiff and will be easier bended towards the tower with wind load on it. Thus as a first guess,

we assume that the blade bending is strongly related to the wind speed. In consequence we try to find now relations between $Z_{H,max}$ and $\bar{U}$ by still keeping the blade pitch in mind. The information of all three variables is combined in the scatter plot of Fig. 15. The WT sailing mode with blade pitch angles close to $70°$ happens mostly when the average wind speed is basically below the cut-in wind speed of $3\,\mathrm{m\,s^{-1}}$, when the power production is not economic or even possible. Clusters of scatter points at high $Z_{H,max}$ are more likely accompanied by $\theta$ values of $90°$, when the average wind velocity is high. The fact that the

turbine is still not going into normal operation during the high wind speed cases might be related to too strong and changing wind gusts making a stable electricity production difficult. For those cases we find also a highly reduced variability of $Z_{H,max}$.

When the WT is in normal mode (grey scatter points) no obvious linear relation between $Z_{H,max}$ and the wind speed is visible. In order to not overload the information given in the plot, the color coded blade pitch dimension is only shown for the abnormal WT modes ($r_s < 1\,\mathrm{rpm}$). The tendency for higher $Z_{H,max}$ with increasing $\bar{U}$ is perceivable and adds up to the

425 positive correlation with the change in WT orientation $\Psi$.

**Table 6.** OLS (ordinary least squares) regression results for fitting $Z_{H,max}$ by the independent variables $\theta$ and $\Psi$. The results are shown for the normal WT operation with rotor speeds $r_s \geq 1\,\mathrm{rpm}$.

| Parameter | $r_s \geq 1\,\mathrm{rpm}$ |
|---|---|
| Dependent variable | $Z_{H,max}$ |
| No. Observations | 2406 |
| Df Residuals | 2404 |
| Df Model | 2 |
| R-squared | 0.914 |
| Adj. R-squared | 0.914 |
| F-statistic | 1.279e+04 |
| Prob. F-statistic | 0.00 |
| Coef. $\theta$ | 0.4065 |
| Coef. $\Psi$ | 73.7703 |

In conclusion, complex interactions between $\theta$, $\Psi$ and $\bar{U}$ should explain to a large extent the occurrence of the maximum returns as well as the overall captured variability. With this knowledge a statistical methods like OLS (ordinary least square) fits can be deployed to the data as introduced in the next Sect. 3.3.

### 3.3 Ordinary least square fitting to explain $Z_{H,max}$ variability

Here we briefly present the outcome of a ordinary least square model fit for the dependent variable $Z_{H,max}$ (2406 data samples) by using the independent wind turbine parameters $\theta$ and $\Psi$ (Pearson $r = -0.0051$) during normal WT operation ($r_s \geq 1\,\mathrm{rpm}$; Fig. 14b). The method should not be applied during the abnormal operation as $\theta$ and $\Psi$ have a linear correlation (s. Fig. 14a) and cannot be handled as independent variables anymore. For the computation we make use of the Python package *scipy.stats* and for more details, also on OLS, we refer to Seabold and Perktold (2010).

The key summary of the OLS model fit results is shown in Table 6. The coefficient of determination, denoted as R-squared, is the proportion of the variance in the dependent variable that is predictable from the explanatory variables. The general use of an adjusted R-squared value is the attempt to account for the phenomenon that R-squared can automatically and spuriously increase when extra explanatory variables are added to the model. In our case there is no difference for both R-squared values, which reach in case of $Z_{H,max}$ 0.914, meaning that over $90\,\%$ of the observed variance can be explained by the parameters $\theta$

and $\Psi$. Also with the fact that the p-value of the F-statistic is less than the significance level, our sample data provides sufficient evidence to conclude that the used regression model fits the data better than a model with no independent parameters.

The OLS model can now be used to rebuild the reflectivity time series $Z_{H,max,ols}$ from the independent variables $\theta$ and $\Psi$. A simple comparison between the original and rebuilded time series for $r_s \geq 1$ is provided in Figure 16. Time periods of good

agreements between the violet and black scatter points can be identified besides those with large differences. With the help of machine learning algorithms in the future, better results are likely to be achieved.

## 4 Summary and conclusions

The interaction between wind turbines and radar systems is complex in nature due to a large set of parameters affecting the scattering behavior of the turbine object. In this context, MeteoSwiss was in charge of organizing two measurement campaigns, lasting 23 (2019) and 22 (2020) days each, with a mobile X-band weather radar (METEOR 50DX) in the proximity of the city of Schaffhausen (Switzerland). The area of a small wind park consisting of three large wind turbines was the target area for the radar measurements. The radar was located at a distance between $7.7$ and $8.6\,\mathrm{km}$ away from the individual wind turbines. There was a fairly good visibility towards the three wind turbines. The measurements taken during March 2019 were close to a receiver saturation, while during March 2020 and the fixed-pointing radar observations with highest returns reaching $78.5\,\mathrm{dBZ}$ a saturation was more likely to happen. Indeed, the observed discrepancy between the correlations of $Z_{H,min}/Z_{H,max}$ and the blade pitch angle $\theta$, for which $Z_{H,min}$ is much higher downhill correlated then $Z_{H,max}$ is uphill correlated, could be attributed to a receiver saturation issue. So to say, the receiver was not anymore able to capture the highest intrinsic returns from the wind turbine.

At the distance of the wind turbines, the beam size was large enough ($\sim 200\,\mathrm{m}$) to entirely include the complex obstacle within the beam shape. Measurement data have been collected with a scanning sequence composed of slowly moving PPI and RHI scans, with a repetition time of $45$ minutes in March 2019 and with a fixed-pointing antenna observation in March 2020. Although the radar data included polarimetric moments and power spectra, the focus of this paper, was the statistical analyses of horizontal reflectivity $Z_H$ and radar cross section RCS of large wind turbines for a weather radar operated within the X-band. The maximum returns are describing a general worst case scenario, which is of interest, for example, to the mostly civilian aviation safety sector when measurements from K- or X-band SMR (surface movement radars) or PAR (precision approach radars) are validated. For the military aviation sector it would also concern radar systems for guidance and surveillance, both for ground-based and airborne systems.

From a comparison point of view, it is worth to look at computational results from Angulo et al. (2015). They also report RCS values on the order of $40\,\mathrm{dBsm}$, with extremely narrow peaks up to $45$ to $55\,\mathrm{dBsm}$, for angles of incidence of about $2.5°$. For other angles of incidence large decreases between $15$ to $30\,\mathrm{dB}$ resulted. Those simulated values do not excessively deviate from our retrieved results. According to those simulations, peak RCSs occur at extremely precise angles of incidence and the worst case situations actually can be observed at slightly negative incidence angles (Angulo et al., 2015), a setup that could not be realized with the presented measurement campaigns. In those conditions, for wind turbines of similar size with respect to the ones shown in this report, RCS values can rapidly increase to more than $60\,\mathrm{dBsm}$. For future campaigns it may thus be interesting to perform measurements in a site that allows observing wind turbines at negative incidence angles. With the high temporal continuous staring observations up to $44.1\,\mathrm{dBsm}$ were reached in case of the closest wind turbine (WT1) with the best visibility from the radar location.

The pattern of the secondary returns depends on the full characteristics of the antenna pattern in all the possible planes, but it is relevant to mention that secondary (or tertiary peaks) have been measured with intensities 20–30 dB lower than the median returns during the measurement campaign in 2019.

Although the derivation of wind turbine RCS is possible by using CEM (commercial computational electromagnetic) tools as proven by Danoon and Brown (2013), the computational requirements are huge and accounting for the incidence angle, radar range, the rotation of the blades and the yaw angle is difficult and requires very large lookup tables. Thus dedicated measurement campaigns with e.g. mobile radars offer another approach to assess wind turbine reflectivity and RCS in a broad range of real environmental scenarios. Bredemeyer et al. (2019) used an UAS (unmanned aerial system) as a passive bistatic radar (PBR) at nacelle altitude for reflectivity measurements of a wind turbine in the C-band (5.64 GHz). From short distances below 300 m they got RCS differences of more than 30 dB between WT rotor planes perpendicular and parallel to the PBR.

In our study, enhancements of the median $Z_H$ could be also related to the WT nacelle orientation, with the highest values being observed when the wind turbine is facing away with respect to the radar location. Interestingly, the $99^{th}$ percentile of $Z_H$ showed also enhancements towards 80–90° and 260–270° (sideways facing towards radar). The reason could not be quantitatively deduced. A combination of an increase of the effective nacelle area (elongated shape) and change in the blade surface due to yawing are potential sources.

A short summary of the key results from the correlation and OLS analyses between the WT and $Z_H$ data sets at the 10 min time resolution are given below:

- The maximum returns increase as the blade pitch angle $\theta$ increases, and this is especially true when the wind turbine rotors rotate slowly (sailing or aero-breaking).

- High variability of $Z_{H,min}$ observed, a bit less for $Z_{H,max}$, when the blade pitch is at 70° or 90° ($r_s < 1$ rpm).

- Positive correlation of $Z_{H,max}$ and negative correlation of $Z_{H,min}$ resulted, when the turbine was in normal (power production) operation mode ($r_s \geq 1$ rpm).

- For high wind speeds, accompanied by aero-breaking pitch angles, we find a highly reduced $Z_{H,max}$ variability compared to sailing WT modes in low wind speeds. Additional, a moderate positive correlation ($Z_{H,max}$ vs. $\bar{U}$) is present for $r_s$ below 1 rpm, which could be related to the aero-elastic bending of the blades.

- By only taking into account the blade pitch angle $\theta$ and normalized relative WT orientation parameter $\Psi = |\sin(\alpha)|$, where $\alpha$ corresponds to the relative orientation, an ordinary least square (OLS) model fit shows that more than 90 % of the $Z_{H,max}$ variance can be explained.

Our results can be helpful for wind turbine interference mitigation measures on radar systems in the future. The present detailed description and analysis has been based on the first meteorological quantity measured by weather radar in the history of radar meteorology: the so-called radar reflectivity expressed in dBZ. We plan to complement the present work with spectral (mean radial velocity and spectrum width) and polarimetric signatures (co-polar correlation coefficient, differential phase shift,

differential reflectivity) of the wind turbines. The next step consist of a stratification of the spectral and polarimetric signatures upon a given rotor speed threshold. Indeed, in case of not-moving rotors, the WT spectral signatures are similar to those of a tall metallic tower (Gabella, 2018): very large and very stable co-polar correlation coefficient, small dispersion of the differential phase shift, stable and slowly varying horizontal, vertical and differential reflectivities, null spectral signature. An example of not-varying horizontal reflectivity can be seen in Fig. 8a on March 19. On the contrary, in the case of moving rotors the degree of complexity and interpretation difficulty is similar to what has been shown in the present paper which has focused only on horizontal polarized radar reflectivity.

*Code availability.*  Pyrad, a real-time data processing framework developed by MeteoSwiss, is available via GitHub under https://github.com/MeteoSwiss/pyrad.git or as conda-forge package *pyrad_mch*. Detailed documentation is provided through https://pyrad-mch.readthedocs.io/en/stable/.

*Author contributions.*  **ML** prepared the manuscript, did the main data analyses and was involved to set up the field campaigns. **JFV** was in main charge of the required software developments within Pyrad and helped with the data interpretation. **ZS** had a leading role in the process of finding the best observation site for the radar system and the set up. **MG** scientifically advised the team in all aspects of the field campaigns and radar meteorology. With the help of **RP**, it was possible to get the turbine data from the wind park operator. Further, he advised the team together with **MFB** on all kind of wind turbine aspects. **JG** was, at the time of the measurements, the team lead. He set up the radar and scanning strategies and was involved in the scientific interpretation of the data.

*Competing interests.*  All authors declare to have no competing interests.

*Acknowledgements.*  The presented work has been supported by MeteoSwiss and the Federal Office for Defence Procurement armasuisse within the frame of the RadarV/MalsPlus project. Further on we would like to thank the Hegauwind GmbH & Co. KG Verenafohren, which kindly provided the operational data of the wind turbines.

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

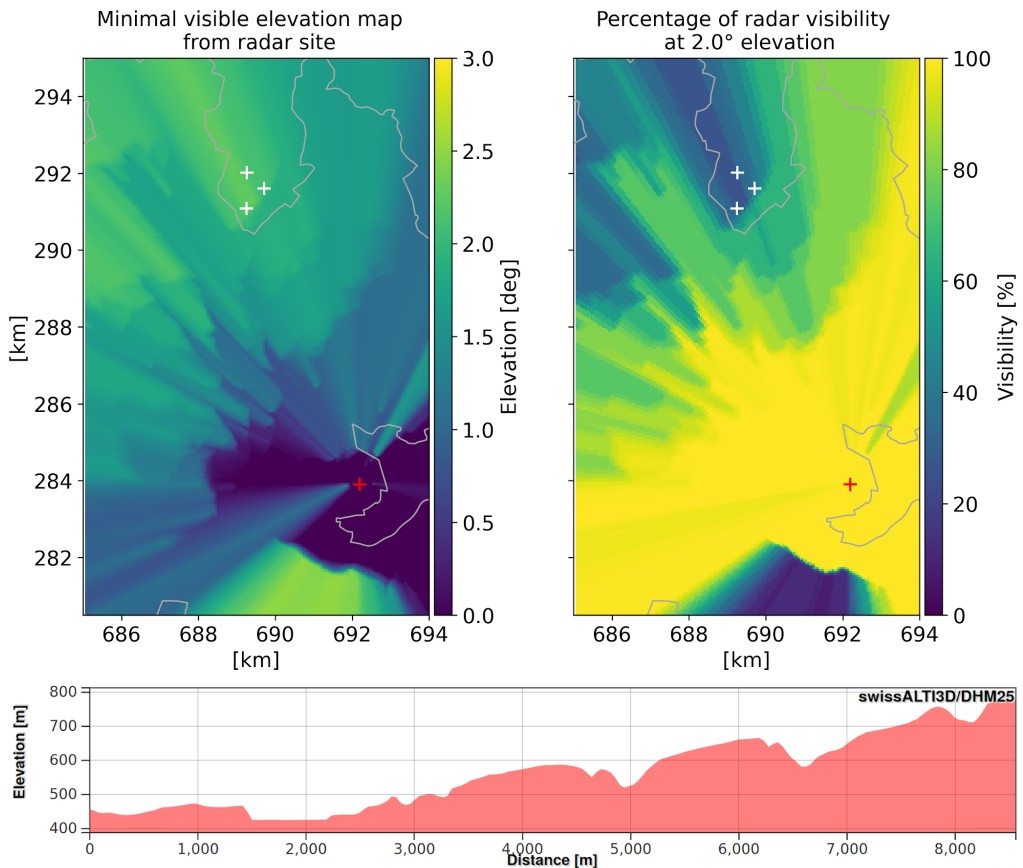

**Figure 1.** Visibility contour maps from DEM based GECSX (Ground Echo Clutter Simulator) simulations for the radar observations site close to Schaffhausen. Minimal visibility map (elevation for maximum of $50\%$ beam blockage) is shown in plot (a). Plot (b) shows the derived percentages of visibility the fixed antenna elevation of $2.0°$). Red plus signs indicate the wind turbine locations and the black plus sign the weather radar location. Additionally the cross section of the terrain profile between the radar observation site (left-end of cross section) and the wind turbine of the wind park with the largest distance to the radar (right-end of cross section) is presented in (c). The last plot was produced on the swisstopo website www.geo.admin.ch.

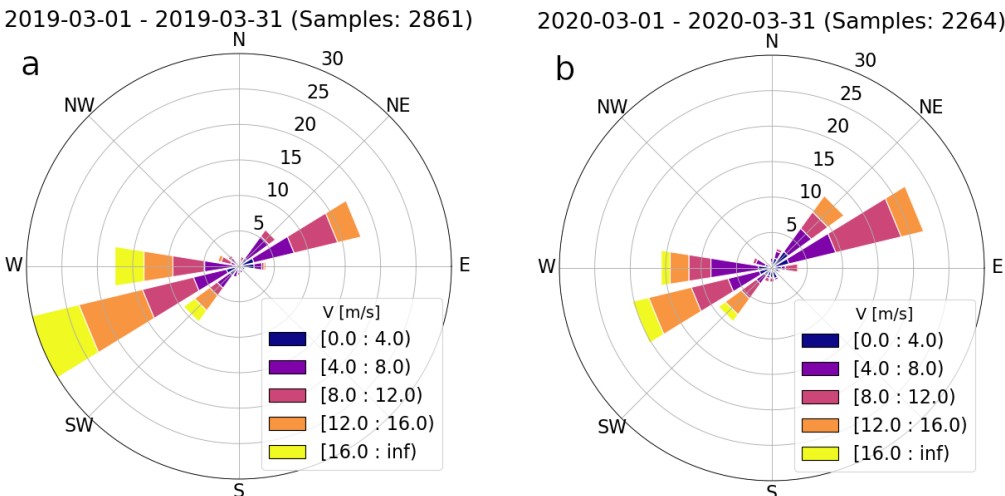

**Figure 2.** Wind rose plots showing the wind speed and direction distribution for March 2019 (a) and 2020 (b) at an altitude of $874\,\mathrm{m}$. The measurement data is retrieved from a nearby wind profiler ($47.69\,\mathrm{N}$, $8.62\,\mathrm{W}$) in the district of Schaffhausen. The altitude of the wind measurements are comparable to the altitude of the nacelles of the wind turbines and the horizontal distance to the turbines is about $8.8\,\mathrm{km}$. The colors indicate the wind speed intervals in a bin width of $4\,\mathrm{m\,s^{-1}}$.

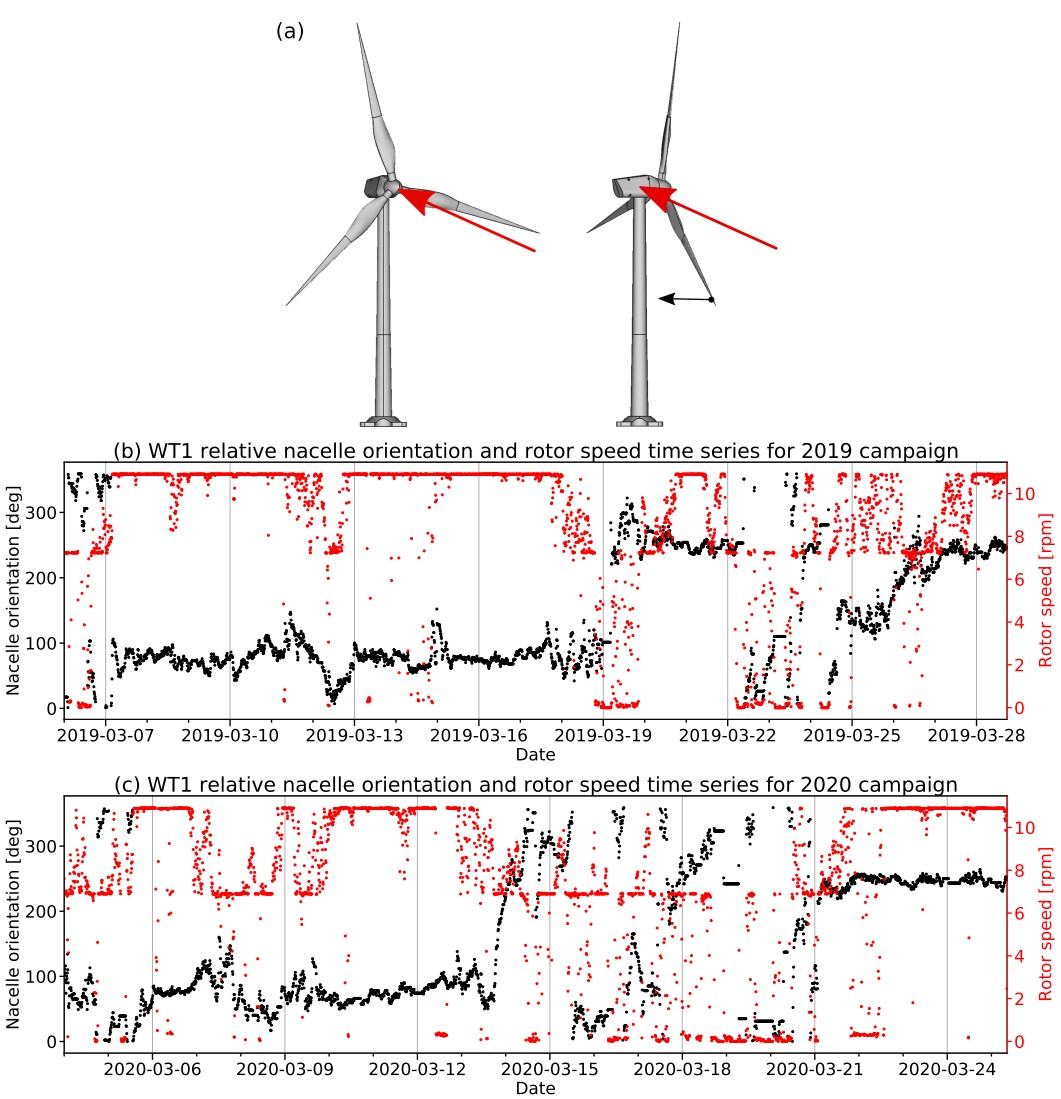

**Figure 3.** In (a) a schematic view of two wind turbines in relative nacelle positions of 0° (left) and 270° (right) with respect to the radar angle of attack (red arrows) are shown. The blade pitch angles are at a high position ($\theta > 70°$) and the direction of blade bending during wind load is indicated by the black arrow. The plots (b) and (c) are showing the time series of relative, with respect to the radar location, orientations (black) of the WT1 nacelle from 2019-03-06 to 2019-03-28 and from 2020-03-04 to 2020-03-25. On top and specified by a secondary y-axis is presented in red the associated rotor speeds in revolutions per minute (rpm). For the measurement campaign in 2019 the time series plot is representatively shown for wind turbine WT1. The offset angle between relative and absolute nacelle orientations is 158.9°.

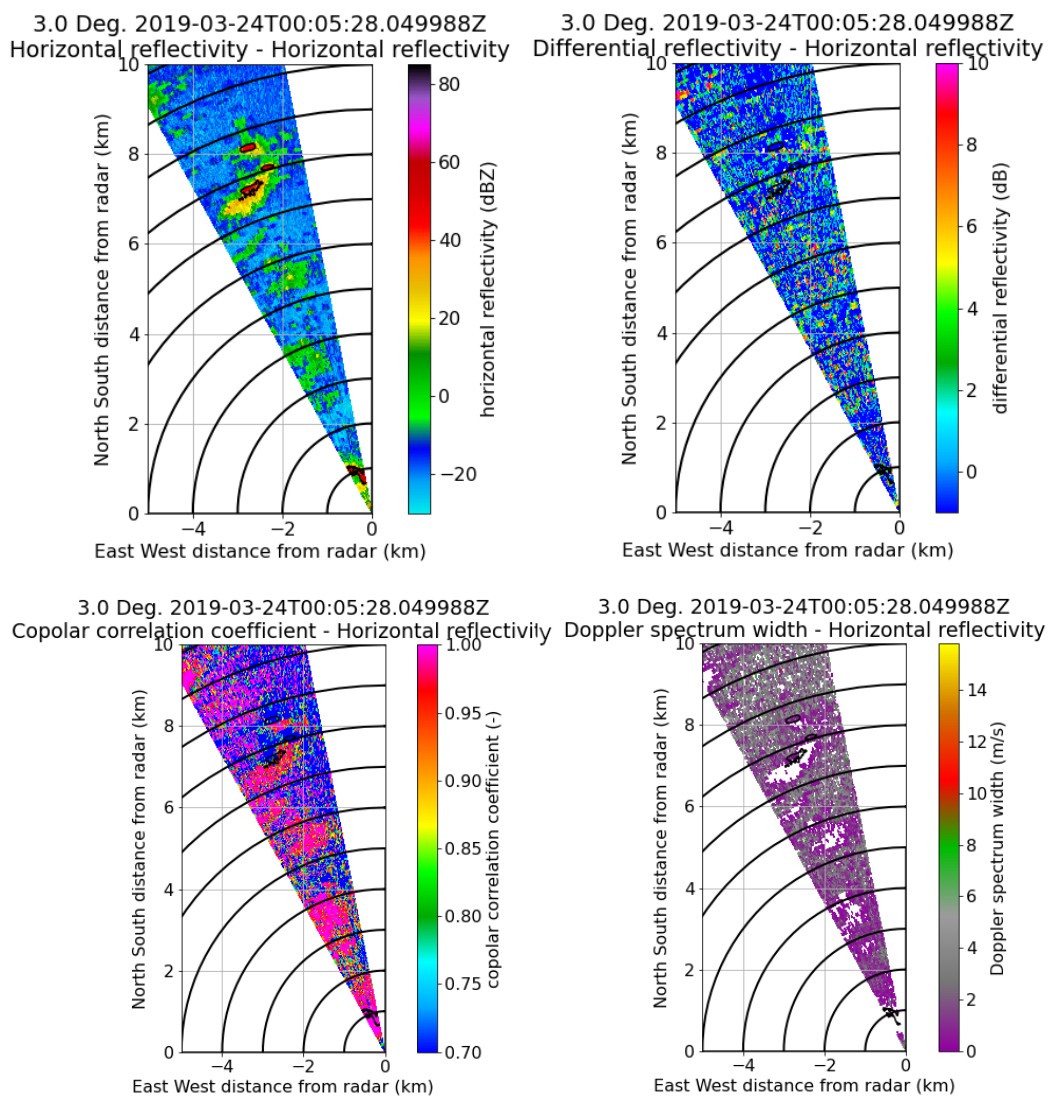

**Figure 4.** Example of single PPI slices (a-d) of horizontal reflectivity $Z_H$, differential reflectivity $Z_{DR}$, co-polar correlation coefficient $\rho_{HV}$ and Doppler spectrum width $W$ covering the wind park of the three wind turbines at an elevation angle of $3°$ on 2019-03-24. The black circles indicate a range spacing of $1\,\mathrm{km}$. In all PPI slices the $35\,\mathrm{dBZ}$ reflectivity contour is drawn to show the approximate location of the 3 wind turbine scatterer.

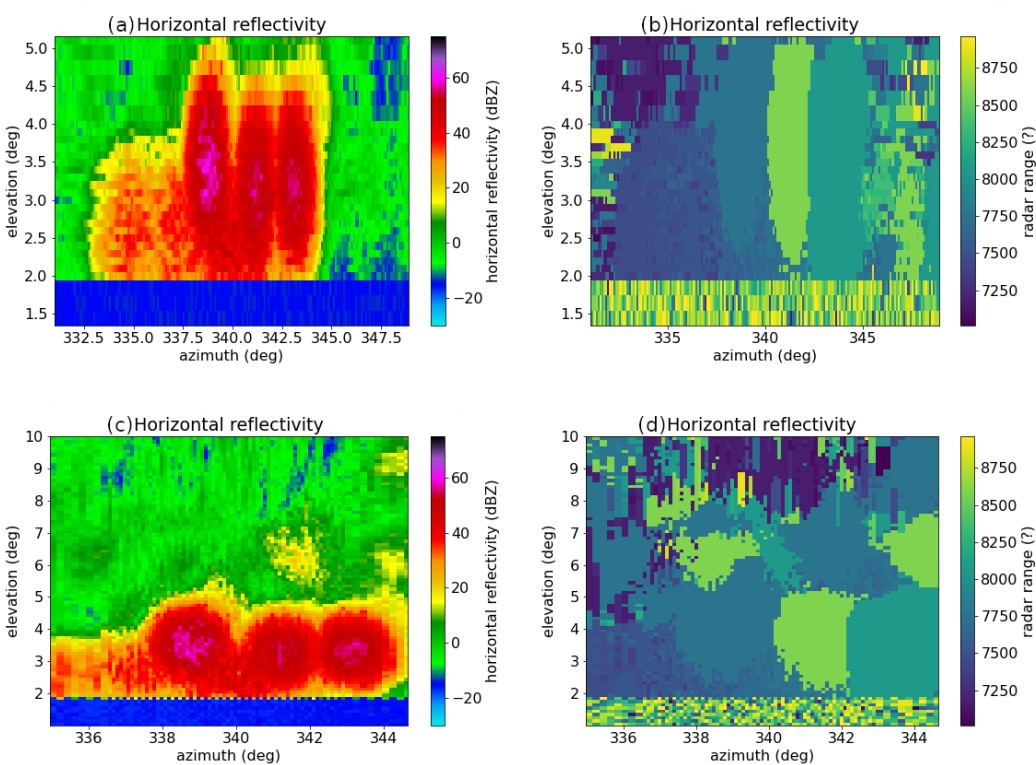

**Figure 5.** Fixed range span plots of the maximum retrieved horizontal reflectivity $Z_H$ from the PPI (a) and RHI (c) scan sequences are shown. The axes indicate the azimuth and elevation angles. The subplots (b) and (d) show the range gates from both scan sequences respectively, where the maximum $Z_H$ occurred. Be aware that azimuth and elevation span is different for the PPI and RHI-based fixed range span plots. The former go from 331° to 349° in azimuth and from 2° to 5.1° in elevation and the latter from 335° to 344.6° in azimuth and from 2° to 10° in elevation.

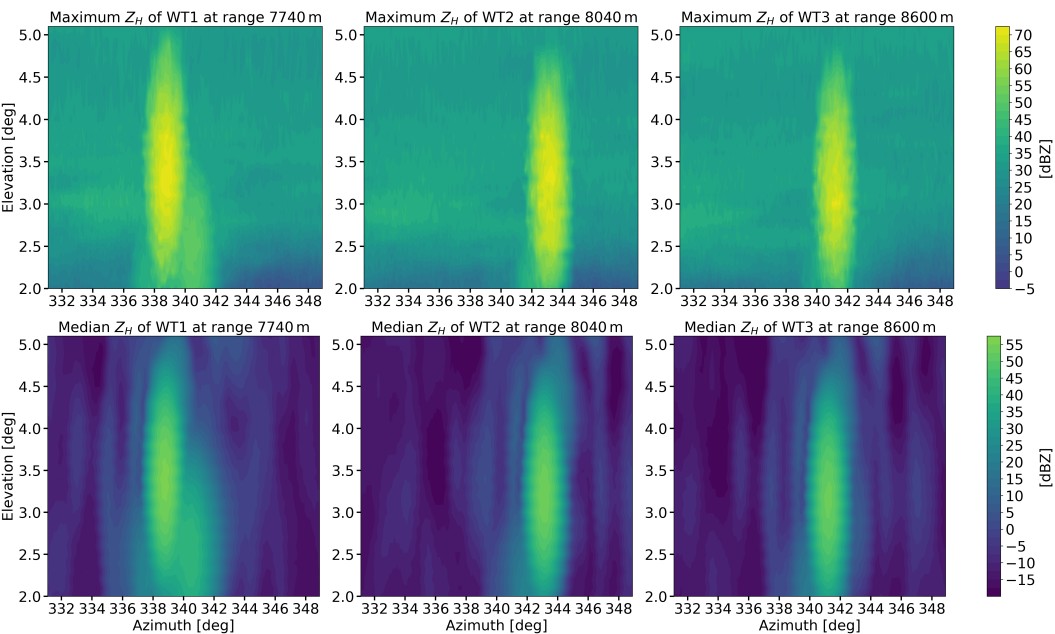

**Figure 6.** Maximum (1$^{st}$ row) and median (2$^{nd}$ row) fixed range statistics from PPI radar scan data of horizontal reflectivity $Z_H$ for all three observed wind turbines during the entire measurement campaign in March 2019. The first column shows the results for WT1 (range: 7740 m), the second column the results for WT2 (range: 8040 m) and the third column the results for WT3 (range: 8600 m). The elevation angles go up to $5.1°$ and the azimuth range is $18°$ (s. Tab. 3).

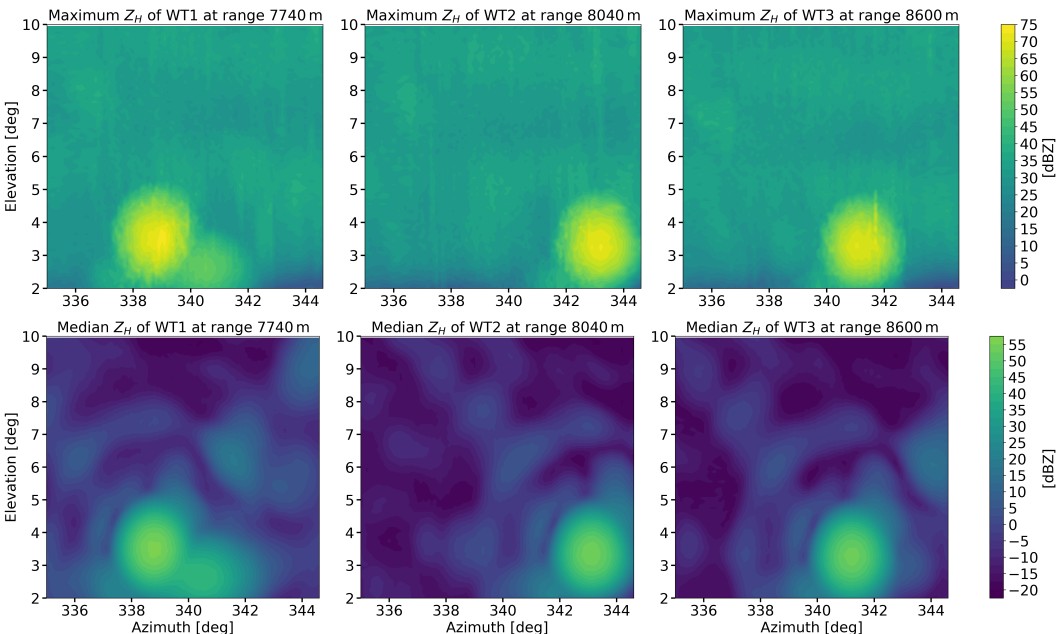

**Figure 7.** Maximum (1st row) and median (2nd row) fixed range statistics from RHI radar scan data of horizontal reflectivity $Z_H$ for all three observed wind turbines during the entire measurement campaign in March 2019. The first column shows the results for WT1 (range: 7740 m), the second column the results for WT2 (range: 8040 m) and the third column the results for WT3 (range: 8600 m). The elevation angles go up to 10° and the azimuth range is 9.6° (s. Tab. 3).

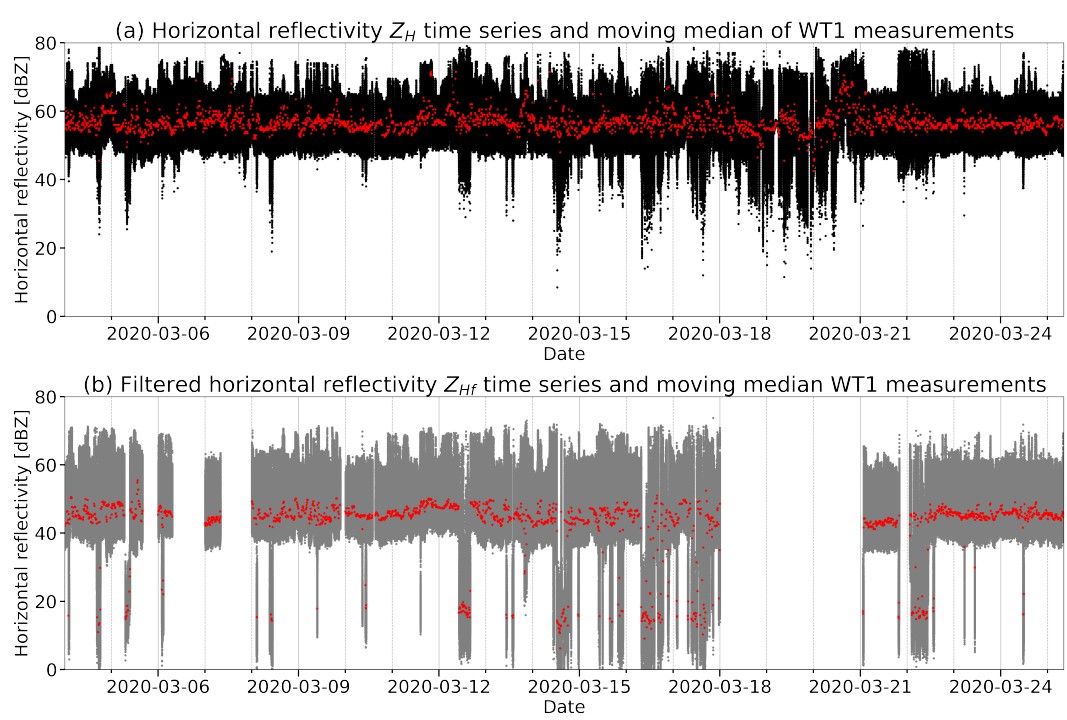

**Figure 8.** In (a), the scatter plot time series (black dots) of horizontal reflectivity $Z_H$ values retrieved from wind turbine WT1 during the stare-mode measurements of the mobile weather radar in March 2020 (2020-03-04 to 2020-03-25). The red colored line represents the corresponding moving median of the reflectivity data with a temporal window size of 10 minutes. In (b), the scatter plot time series (grey dots) of the clutter notch Doppler filtered $Z_{Hf}$ values and corresponding moving median. The clutter filter used has a width of $7\,\mathrm{m\,s}{-1}$ and was applied during postprocessing to moment data where spectral data have been available. In consequence gaps appear in the time series of $Z_{Hf}$.

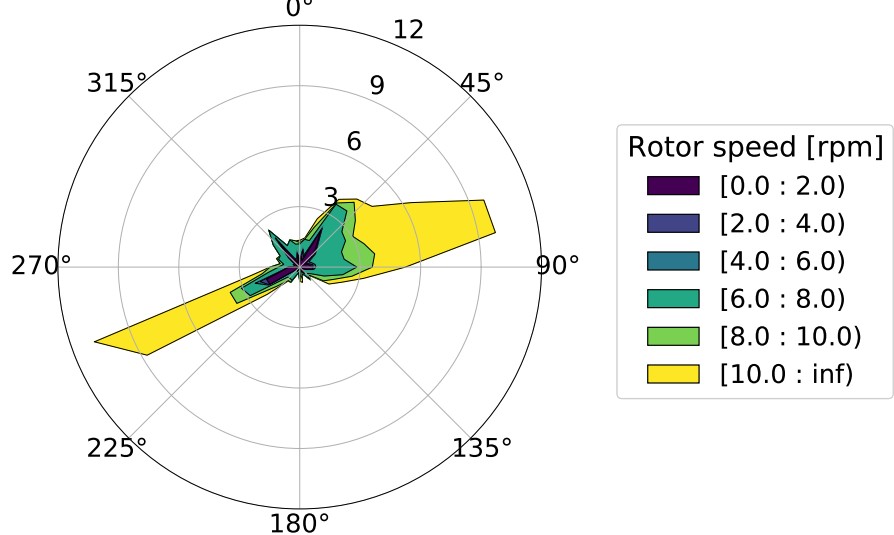

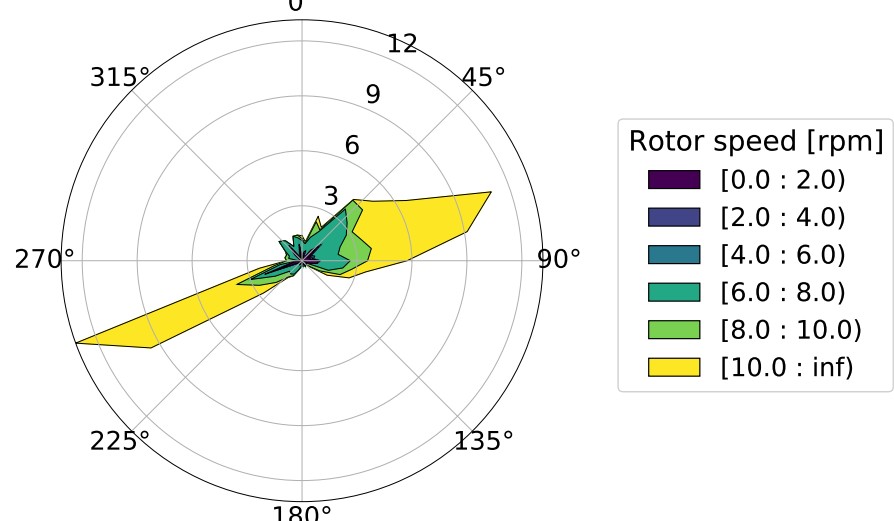

**Figure 9.** The plots indicate the normalized distribution of relative nacelle positions with corresponding rotor speeds during March 2020 for the unfiltered $Z_H$ (a) and filtered $Z_{Hf}$ (b) reflectivity data. The rotor speeds are binned with a width of $2\,\mathrm{rpm}$.

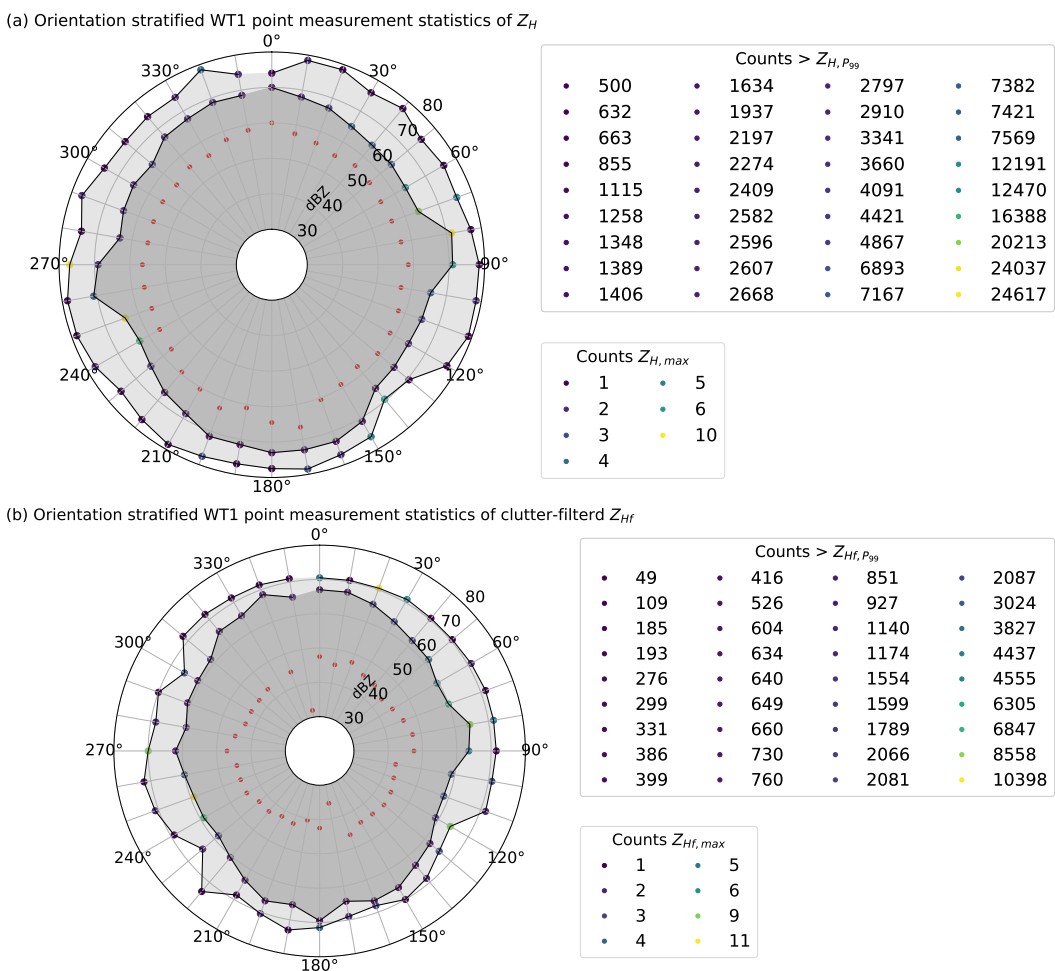

**Figure 10.** Fixed-pointing measurement statistics of horizontal reflectivity $Z_H$ (a) and clutter-filtered $Z_{Hf}$ (b) based on the relative position of the WT1 nacelle. Indicated by the red scatter points in all polar projections is the median value. The maximum values with the corresponding counts within a bin width of $0.5\,\text{dB}$ are shown by the light grey shaded area and color-coded scatter points. The dark shaded area represents the results for the $99^{th}$ percentiles with the counts exceeding them indicated by the corresponding color table.

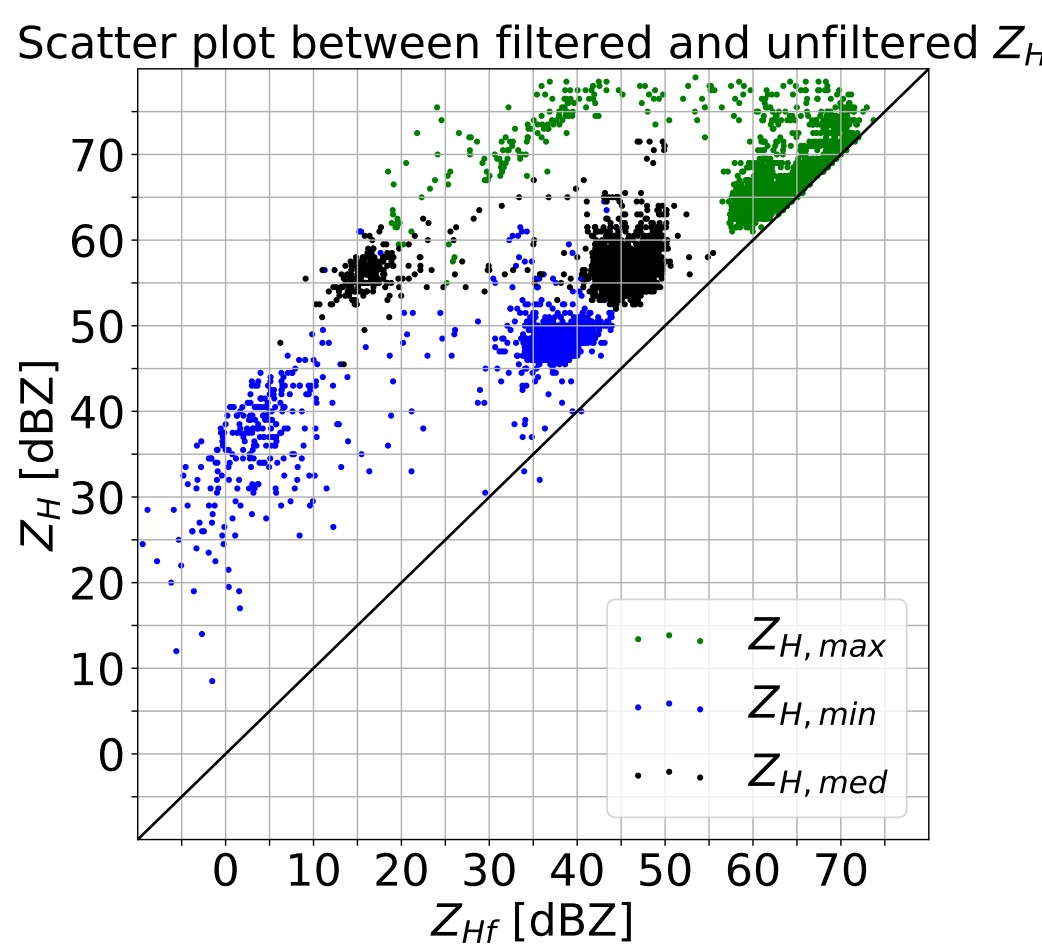

**Figure 11.** Scatter plot comparison of 10 minutes moving minimum (blue), maximum (green) and median (black) $Z_{Hf}$ from Doppler filtered (clutter notch of $\pm 3.5\,\mathrm{m\,s^{-1}}$) and unfiltered reflectivity data $Z_H$.

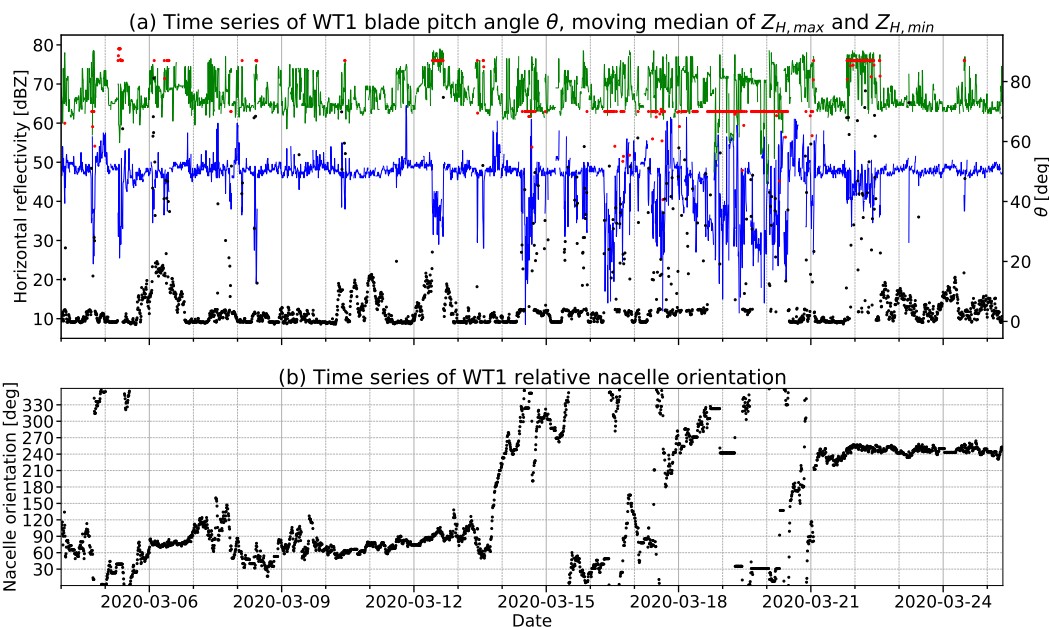

**Figure 12.** The line plots in (a) show the $10\,\mathrm{min}$ moving maximum (green) and minimum (blue) of $[Z_H] = dBZ$ as presented in Fig. 8. The black ($r_s \geq 1\,\mathrm{rpm}$) and red ($r_s < 1\,\mathrm{rpm}$) scatter plots indicate the WT1 blade angles with a temporal resolution of $10\,\mathrm{min}$. The time period between 2020-03-04 and 2020-03-25 covers the whole 2020 field campaign. For practical reasons plot (b) presents the relative nacelle orientation accordingly to plot (a).

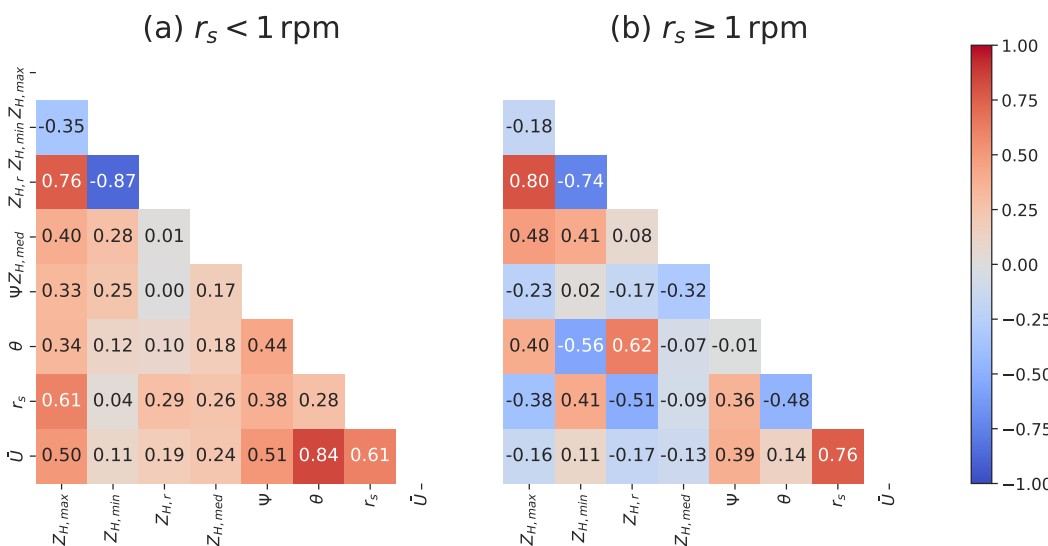

**Figure 13.** Pearson linear correlation coefficients heat map matrices between radar returns re-sampled to a temporal resolution of $10\,\text{min}$ (maximum $Z_{H,max}$, minimum $Z_{H,min}$, difference $Z_{H,r}$, median $Z_{H,med}$) and WT1 orientation $\Psi$, blade pitch angle $\theta$, rotor speed $r_s$ and average wind speed $\bar{U}$. The correlation matrix (a) is computed with the measurements when the rotor speed $r_s$ is less than $1\,\text{rpm}$ and (b) when $r_s$ is equal or higher than $1\,\text{rpm}$. Some corresponding regression line fits are shown in the scatter plots of Fig. 14.

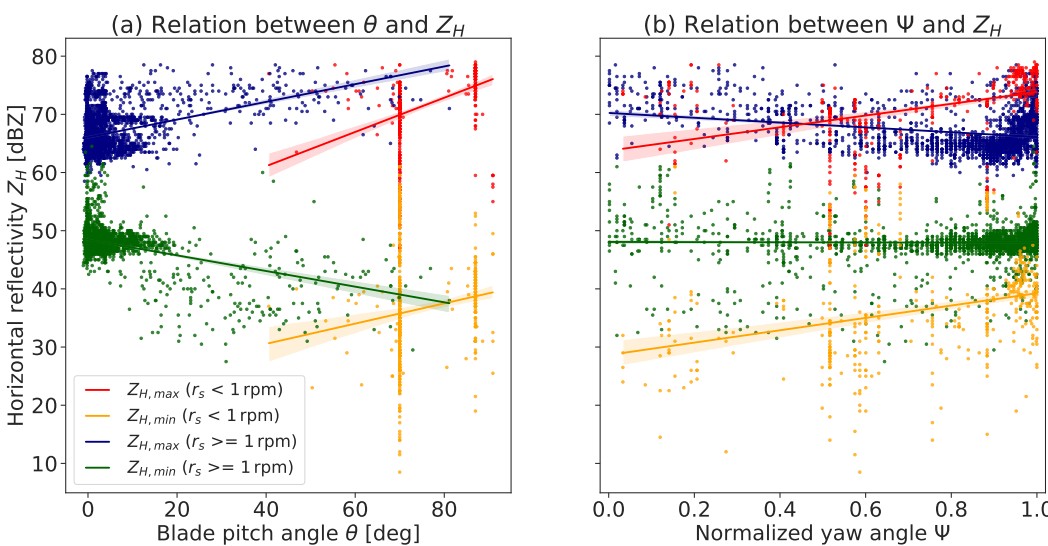

**Figure 14.** Scatter plots and corresponding linear regression lines with confidence interval sizes of $95\%$ for the regression estimate (translucent bands) showing the relations between the blade pitch angle $\theta$ (a), respectively normalized nacelle orientation $\Psi$ (b), and different sub-data sets of $Z_H$. In particular the data set $Z_{H,max}$ (blue and red) and $Z_{H,min}$ (orange and green) are each separated for the WT1 rotor speed threshold of $1\,\mathrm{rpm}$. The confidence intervals are estimated using a bootstrap method.

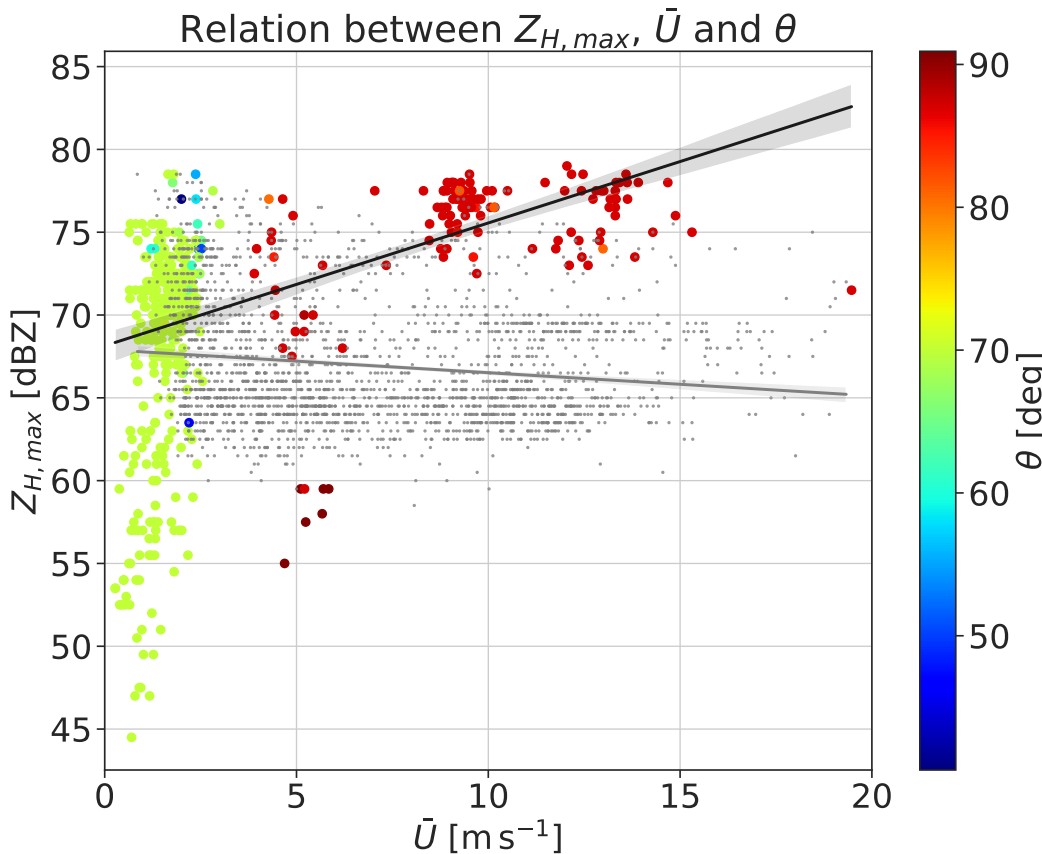

**Figure 15.** Scatter plots and corresponding linear regression lines with confidence interval sizes of $95\,\%$ for the regression estimate (translucent bands) showing the relations between the average wind speed $\bar{U}$ and two sub-data sets of $Z_{H,max}$. The data when the rotor speed $r_s \geq 1\,\mathrm{rpm}$ is plotted in grey, while the data for $r_s < 1\,\mathrm{rpm}$ is represented by the color-mapped scatter points. The color dimension shows the blade pitch angle $\theta$. The black regression line belongs to the color-mapped scatter points. The confidence intervals are estimated using a bootstrap method.

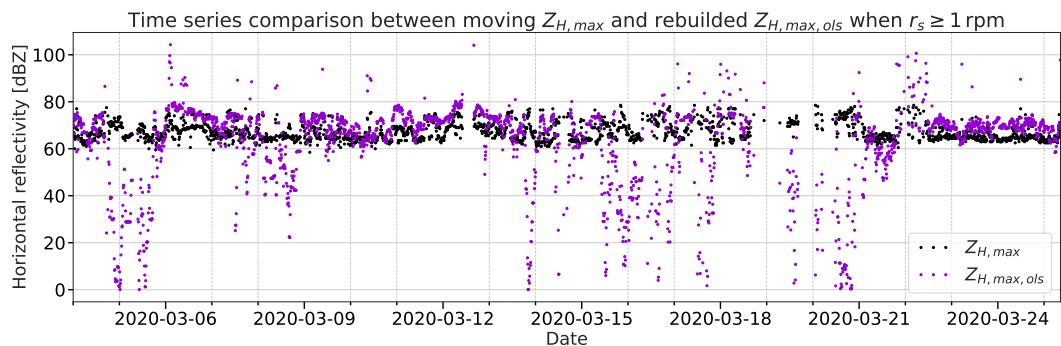

**Figure 16.** Time series scatter plots of the $10\,\mathrm{min}$ moving maximum of $Z_{H,max}$ (black) and the corresponding OLS model based rebuilded maximum of the horizontal reflectivity $Z_{H,max,ols}$ (violet) when the rotor speed $r_s \geq 1\,\mathrm{rpm}$. For the input of the model estimation only the WT blade pitch angle $\theta$ and nacelle orientation $\Psi$ are used.