# Peer review of "Insights into wind turbine reflectivity and RCS and their variability using X-band weather radar observations"

_Atmospheric Measurement Techniques, 2020_

## Referee Comment (RC1) · Anonymous Referee #1 · 7 Dec 2020

Review on

Insights into wind turbine reflectivity and RCS and their variability using X-Band weather radar observations by Martin Leiner et al.

Overview:

For a long time I did not review a manuscript that was as interesting and gave me as much pleasure as this one. The paper discusses a relevant topic, it is clearly structured and comprehensive, most of it could be published as it is. Thank you to the authors and congratulations.
[Figure]

Disturbing backscatter from wind turbines (WT) is discussed fiercely in the radar meteorology community. Getting insights into the properties of wind turbine echoes is the needed basis to develop methods to alleviate or overcome the problems. The authors discuss widely the echo intensity of WT echoes based on target-oriented (in double sense) scanning strategies.

As I said, this contribution is helpful and interesting as it is. Nevertheless, the measurements as they are described provide further information about the echoes from locations nearby around the WT and on the properties of other variables (spectral width, polarimetric variables) at the WT location and around. This information is urgently needed to comprehend the WT problem and I want to encourage the authors to add further publications based on the data of this experiment.

Major remarks:

As I did not find differing information I assume (I have to assume) the given data are measured without a Doppler Filter and any other quality filter applied. It might be I forgot that this information is given somewhere. If not, it has to be clearly stated how the data are processed (by the radar).

Furthermore, for practical application of the results the differences between uncorrected reflectivity (uZh) and Dopper corrected reflectivity (Zh) need to be shown and discussed. The guessing about the importance of the (mainly static) nacelle of the WT for the echoes is not necessary as a Doppler filter will separate echoes from the static parts of the WT from rotor echoes.

The first part of chapter 3 introduces some theoretical terms, especially radar cross section (RCS). It starts from the standard discussion of an single, isolated target. Equation (1) is valid for an particle on the radar axis. If its position deviates from the main axis, the directivity pattern of the radar antenna has to be taken into account.

A WT is not small compared to the 3dB beamwidth. Here the limitations start. The RCS

as it is derived here _is_ sensitive to the directivity patten of the antenna. A different pattern would produce a different weighting of individual parts of the WT, resulting in a different RCS. Additionally, the WT is not small compared to the radial resolution of the measurements (75 m). Already a flagpole (i.e. a point in a PPI) contributes to two consecutive range gates if it is not perfectly positioned at the center of one range gate. As the diameter of the antenna is nearly twice the length of a range gate, the echo from the WT is distributed in three or four range gates if the relative yaw angle is 90 degrees or close to it. One range gate never contains all the echo from the WT. This needs to be explained and taken into account when deriving RCS from reflectivity.

In the following text and the figures the authors provide sometimes reflectivities, sometimes RCS values. As stated above, I do not trust the RCS and I do not want to calculate back an forth to compare RCS and Z values. My recommendation is: (i) improve the derivation of RCS by discussing the effect of the spatial extend of the WT, (ii) determine the offset between RCS and Z for each of the WTs, and then (iii) stay to the measured reflectivities without introducing additional uncertainties from the conversion to RCS.

I do not follow the argument of the orientation of the nacelle to be important for the backscatter (line 288 and later). (i) The (scattering) area of the nacelle is small compared to the area of the rotor. (ii) The nacelle often has more or less straight "walls", leading to a stealth behaviour: as long as the radar is not perpendicular to the nacelle surface, the (strong) echo is not scattered back to the radar. (iii) The scatter properties of a nacelle might be severely dependent on corners on the surface of the nacelle, which can have the effect of a corner reflector. Thus the properties of this individual nacelle are not representative for other type of WTs.

In order to prove the theory of strong nacelle echoes, the authors should discuss the effect of a Doppler filter. Such a filter should be able to suppress the nacelle echo, as the nacelle is (nearly) not moving.

[Figure]

The last concern is about the linear correlation coefficients used in Fig. 11 and the text on it. In case there is a relation between WT physics and radar echoes it is not necessarily linear but probably strong. Assuming a linear relation between pitch angle and echo strength means - just to clarify my argument - a significant stronger echo at 390 degrees pitch angle than at 30 degrees. The surfaces on the rotor blade are in no way changing in a linear way.

With this argument, taking |sin| of the yaw angle is useful step. The images in Fig 12 bring some evidence that linear relations are not totally misleading. Nevertheless for |sin(yaw angle)| there is a significant shift to stronger echoes, not described by the linear relation.

Fig 11 shows 28 correlation coefficients for three different samples of the measurements. Only 16 of them relate a WT related value (psi, theta, rs, and u) with a radar variable (Zmax, Zmin, delta Z, Zmedium). Wouldn't it be possible to show more of these scatter plots but only the four relations for two different samples?

In Figs 12 and 13 there are 95 % confidence intervals marked by shaded areas. But in no case 95 % of the measurements fall into these intervals. What is the meaning of these 95 % confidence intervals?

Lately, the values for Zhmax are limited by saturation of the receiver as the authors mention. What does this mean for correlation coefficients of Zhmax to some other value?

Minor remarks:

L 23: 205 GW is the _installed_ capacity of WTs. As the wind is unsteadily blowing, the amount of yearly produced energy is significantly less than 205 GW times 8760 hours (=1.8 PWh). The real production was "only" 417 TWh. These 417 TWh correspond to 15 % of the electricity consumption of Europe. Do not mix up these terms. BTW: There is no proof that the number of wind turbines needs to increase. It might be the size of

the turbines instead.

L 93: The directions towards the WT from the radar are 337.8°, 343.3°, and 340.2°. The RHI scans (according to Tab. 3) are only performed until 342.9° in fine resolution. Why does the range not cover the turbine at 343.3°? In Fig. 4d the positions of the most intense echoes are at roughly 338.5°, 341.5° and 343.3° which does not fit to the (first two) given WT locations. Same discrepancy for Fig. 5 and Tab. 4 also gives a different location of WT 1.

L 149: Sorry, I do not "clearly" see low ZDR and high RhoHV values at the wind turbine positions. Could you add an isoline of reflectivity at 35 dBZ to make it easier to see the effect? ZDR and RhoHV show an intense variation (noise) making it hard to assign individual pixels to a WT effect.

L 177: The inline equation contains two different Pr. One has the unit mW, one the non-unit dBm. The equation enables us to determine Pr. I know the problem to introduce dB and values on a logarithmic scale, but this way is wrong.

L 179: sigma is not described as a scalar but deeply discussed as a function of the two angles occurring. What is a scalar is sigma(pi), the _back_scatter cross section.

L 180: I added a linefeed before "Precipitation".

L 192: In equation 4 the gain G0 looses its 0 without further notice. Why?

L 198: if you change equation 5 to sigma_i = pi^5 |K|^2 D_i^6/lambda^4, there is no need to change from sigma_d to sigma_i.

L 201: You did not introduce Z, so you cannot convert from Z to RCS.

L 204: You should give all three conversion factors for the three WT.

L 248: "Averaged value"? How did you average? In dB or in m^2?

L 248: The difference between two logarithmic values is always dB, not dBsm.

L 280: Yes. The yaw angle of the WT is independent of reflectivity values. It is even independent of the presence of the radar. The message should be, the maximum reflectivities are insensitive to the yaw angle. - BTW: 10 dB difference is not really insensitive. It might be less intense than expected, but 10 dB is a factor 10.

L 285: The variation of the maxima (in a) is in the order of 10 dB. The variation of the P99 values is also 10 dB. Why do you say, a directivity is more visible in b than in a? Same question occurs for the median values. - Plot (just for you) the figures into a Cartesian coordinate system and see if the messages is stable.

L 417: See remark on line 248.

Figures:

The figures are a main drawback of this paper. It is the duty of the authors to provide figures the show - not hide - the information. Along all the figures only a few axis descriptions, titles and so on are readable. I do not want to use a magnifying glass to read the figures! Hardly any of these figures can be accepted as it is.

Additionally, figure 12 makes use of bright and dark green and blue. It might be that the printer of the authors (or their screen) shows these differences. Mine not. There are colors which are better distinguishable than bright and dark blue. There are different marks but + signs.

Fig 1. The black + for the radar position is totally invisible in my printout in subfigure a. Same for the border lines in both figures. Axis labels and tic labels are fine for a and b, tic marks for c are at the limit.

Fig 2. Absolutely unreadable in printout. On the screen at 400 % size acceptable but the velocities at the color codes.

Fig 3: Axes labels and tic labels at the limit. Sketch of wind turbines is a good try to explain nacelle orientations. I think, a view from the top is easier to see.

[Figure]

There is no additional information but only more confusion in showing absolute and relative orientations of the WT. Simply give the offset angle and show one of them.

What is the source of the echoes left of WT 1 in d and e?

Fig 4: Can you provide a PPI of spectral width as well? Labels in a through c at the limit, in d through f totally ridiculous.

Fig 5 and 6: If you assess the patterns around the wind turbine to the directivity pattern of the radar antenna, we need that directivity pattern to be shown in the same form. Is there any difference between the top rows and the bottom rows but an offset? If not, bottom rows are not needed.

Axis and tip labels are below the limit.

Fig 7: In a, c and f the local maximum at the position of the WT is a bit larger then the overall maximum around the WT. This is not possible by definition.

Axis, tip and legend labels are totally unreadable. Digitally they appear to be readable at roughly 250 %.

Fig 7/8: One figure shows RCS in dBsm, one figure shows reflectivity in dBZ. This is not helpful. Please show always Z in dBZ.

Fig 8: Axis and tip labels at the very limit.

Moving average in dB, not in mmˆ6/mˆ3?!

Fig 9: Labels and so on much too small.

The figures are strongly missleading! Whereas the reflectivity scales show a range of 60 dB, the cross section scales only show 40 dB. This changes the shape massively!. Max reflectivity at $110°$ is roughly 80 dBZ, at $140°$ it is 70 dBZ, 10 dB less. Max RCS at $110°$ is fairly 45 dBsm, at $140°$ it is 35 dBsm, also 10 dB less but the "bulge" looks much stronger - it is not.

The median reflectivity has a difference of 68 dBZ (170°) down to 56 dBZ (310°), i.e. roughly 12 dB. The median RCS varies between 32 dBsm (170°) and 20 dBsm (310°), also 12 dB. All the difference that are visible between a/b and c/d are caused by the way of presentation.

I recommend to combine a and b into one figure (having three dot circles instead of two, showing only values between 80 dBZ (max value of all) and 50 dBZ (min value of median values).

Bin width are differences and thus always given in dB, not in dBZ or dBsm (see comment on line 240).

Fig 10: "WT1 blade angles" means pitch angles?

Fig 11: Do we need more than the lower 4x4 CC values? Probably instead of simple linear correlation coefficients scatter plots could tell more.

Fig 12: Especially the legend is unreadable. Bright and dark green, bright and dark green are indistinguishable in my printout. Choose other colors.

Fig 14: Legend unreadable.
* * *

---

## Referee Comment (RC2) · Jochen Bredemeyer (Referee) · 23 Jan 2021

My overall statement: I find this paper very valuable, since it develops in a comprehensive form the deviation of a single WT's RCS from a a measured dBZ gained from a precipitation radar. Special care is taken to set up a measurement to isolate the radar return from specific wind turbines. Much work is invested to present a meaningful statistical analysis.

Remark on nearly all Figures: The tics and axis names are to small, and some important marks in the diagrams are also too small. This makes some figures quite unreadable.

Detailed comments referring to paragraphs:
________________________________________-

Abstract (Paragraph 1) and Introduction (i.e. Par. 45):

The radar in use is a "precipitation radar". Please use at least once this term.
* * *
Par. 30:

"RCS is an optimal variable to estimate the effect of a wind turbine on the performance of a radar system, in fact existing numerical models for estimating the back-scattering efficiency of wind turbines rely on this quantity."

According to [1] ("scattering cross-section:") the incident field is assumed to be planar over the extent of the target. This is in principle not the case for objects on the ground and causes some restrictions.
* * *
70:

I am missing a hint or a justification why the radar pulse width was selected $0.5\mu s$ but no other values. Is there a technical restriction? Same for PRF.
* * *
125:

"... co-polar correlation coefficient are represented on 2019-03-24."

Are these coeffecients in use later when developing eq. (7) using eq. (3) ? (3) is defined for circular polarization.

"During the fixed-pointing scans, the antenna of the radar was not moving and always pointing to the same wind turbine"

Fig. 2 too small, unreadable

———

150:

"The wind turbine clutter is clearly visible at the range from 7.7 to 8.6km as high ZH, HV and low ZDR and all three turbines can be distinguished from other ground clutter signals"

———

165:

"In the following part of the paper, all the horizontal reflectivity ZH measurements and retrieved RCS data ..."

See restrictions of RCS mentioned above and discuss the applicability.

———

175:

eq. (1) is simply a form of the radar equation and should be referred to as it is.

———

185:

"Accounting for the actual distribution of power within the beam generated by a circular parabolic antenna, a correction factor of $1/(2 \mathring{A}\mathring{u} \ln2)$ was introduced by Probert-Jones (1962):"

The development of the later eq.(7) has some uncertainties, i.e. the usage of eq. (3) which applies to circular polarization. Please mention that and discuss what the restrictions may be applying linear polarization.

———

195:

"The Rayleigh approximation of the backscattering cross section of a single water drop d can be expressed as ..." Is there a reference to literature?

———

215:

"The statistical overviews in Figs. 5 and 6 show the maximum (1st row), median (2nd row) of ZH and the maximum retrieved RCS (3rd row) for all three wind turbines."

So equation (7) is used for that representation, please make a note.

———

250:

"Figure 7 shows the variability of the location and intensity of the RCS maxima. The farther apart the blue and black histograms are, the higher is the variability of the distribution of the maxima."

Please explain this in more detail.

In Figure 7, both curves represent RCS values according x-axis. However, the legend says "Pmax" and "RCSmax", which is confusing. Please correct or clarify.

"The blue histograms are calculated at a fixed position (as listed in Table 13)"

Mistake: Table 13 ?? –> Table 3

———

265:

From the 2020 measurement, it may be interested to zoom much deeper into time to see the fluctuation across a short period, i.e. within some seconds in contrast to the long-term diagram as of Fig. 8.

Figure 9: "dBZ" and "dBsm" refer to absolute values. If the meaning is a variation, i.e. for a bin, always use "dB" alone without reference!

Same mistake as in line 265 "about 4dBsm higher". There are maybe more of these in the entire text. In line 285 however, it's correct: "... bin width of 0.5dB".

——

280:

"From the polar plots in Fig. 9(a, c) we see that the relative orientation of the turbine/nacelle is insensitive to the maximum values, evident by the round and not too much disturbed distributions"

Is this correct for a) _and_ c) ? For a) I agree. I think for the median values (red) this is correct for a) and b) depicting the dBZ. Is the averaging time always the 10 min as of Fig. 8 ? In sub-figures c) and d) the difference of medium RCS across the two positions "radial 90/270°" and "tangential 0/180"" is lower than I would expect.

Could that be validated from other sources/measurements for X-Band (references)? I have seen many RCS simulations for WTs in C band that claimed differences >20dB. From own measurements close to reflecting WTs we got differences of >30dB with rotor planes in frontal (0°) and side (i.e. 90°) view [2]. However, there is obviuosly a difference from the radar's point of view. It observes the target over a larger distance that includes terrain effects.

——

290:

"Future work should take also into account numerical simulations ..." Yes, that might be interesting. I have not seen many in the X band. However, methods like MoM or even MLFMM might not be suitable due to the huge number of triangles to work with. Asymptotic methods I(like PO, GO), however, might neglect some eletromagnetic

(near) field effetcs.

——

300:

"For example on 2020-03-12 or 2020-03-22 blade pitch angles of WT1 were aligned at roughly 90° over hours..."

In Fig. 10 it would be helpful alos to have an additional curve that simultaneously shows the orientation across the time axis.

——

345:

Why is the relative yaw angle Psi normalized, and the other angles are not?

——

410:

Ground radars in X (or K) band are used in aviation only as SMR (surface movement radars) or for the pecision approach radar (PAR). Unfortunately, for surveillance or air defense radars in S band, the actual antenna beam widths are much too large to discriminate single WTs at a far distance. Those measurement results, obtained from a similiar measurement campaign, would be really interesting.

——

425:

"... the computational requirements are huge..."

Yes, and be aware of the electroagnetic solvers methods and their limitations (mentioned above).

——

430:

"Thus dedicated measurement campaigns with e.g. mobile radars offer another approach to assess wind turbine reflectivity and RCS in a broad range of real environmental scenarios"

Yes, this is necessary. As an example, We did that for C band precipitation radar [2].

——

############################################################

[1] IEEE Std 211-1997: IEEE Standard Definitions of Terms for Radio Wave Propagation. https://ieeexplore.ieee.org/servlet/opac?punumber=5697

[2] J. Bredemeyer, K. Schubert et.al.: "Comparison of principles for measuring the reflectivity values from wind turbines", International Radar Symposium - IRS 2019, Ulm, Germany

---

## Author Comment (AC1) · 12 Mar 2021

Martin Lainer (on behalf of all co-authors), 12.03.2021

Color Code: **Referee comments**, **Author responses**, **Relevant changes in the manuscript**

We are highly grateful to the overall positive review and in our view good suggestions.

In the following part we will give point by point answers and/or comments to all points raised by the Referee #1. We try our best to further improve the paper. Relevant changes in the manuscript will be explained or cited with highlighted green color coded text. For simplicity we only provide links towards the sections. For the detailed changes please refer to the visual mark up version produced by *latexdiff*. Be aware that the visual mark up did not work for the Figures and their captions smoothly and also not for the references.
* * *
As I said, this contribution is helpful and interesting as it is. Nevertheless, the measurements as they are described provide further information about the echoes from locations nearby around the WT and on the properties of other variables (spectral width, polarimetric variables) at the WT location and around. This information is urgently needed to comprehend the WT problem and I want to encourage the authors to add further publications based on the data of this experiment.

It is true, that for instance polarimetric echoes are highly interesting to study. A a lot of data has been generated during our campaigns and cannot be presented all in one paper. This first paper shall give an overview about the campaigns and give a first insight into some first promising results. Further scientific studies will follow and build up on this paper.

Section 4: In the end of the manuscript we extend the outlook with this paragraph: "The present detailed description and analysis has been based on the first meteorological quantity measured by weather radar in the history of radar meteorology: the so-called radar reflectivity expressed in dBZ. We plan to complement the present work with spectral (mean radial velocity and spectrum width) and polarimetric signatures (co-polar correlation coefficient, differential phase shift, differential reflectivity) of the wind turbines. The next step consist of a stratification of the spectral and polarimetric signatures upon a given rotor speed threshold. Indeed, in case of not-moving rotors the WT spectral signatures are similar to those of a tall metallic tower (Gabella, 2018): very large and very stable co-polar correlation coefficient, small dispersion of the differential phase shift, stable and slowly varying horizontal, vertical and differential reflectivities, null spectral signature. An example of not-varying horizontal reflectivity can be seen in Fig. 8a on March 19. On the contrary, in the case of moving rotors the degree of complexity and interpretation difficulty is similar to what has been shown in the present paper which has focused only on horizontal polarized radar reflectivity."

Major remarks:
As I did not find differing information I assume (I have to assume) the given data are measured without a Doppler Filter and any other quality filter applied. It might be I forgot that this information is given somewhere. If not, it has to be clearly stated how the data are processed (by the radar).

Your assumption is correct and we forgot to mention it clearly in the manuscript, that our data is measured without any Doppler/quality filter applied. Already in the introduction we will add this important information. Later when we add the analysis of the filtered data, it will get more precise too.

Section 1: " …regarding unfiltered horizontal reflectivity ZH for all three wind turbines during the field campaign in 2019. For simplicity, we call ZH just horizontal reflectivity hereinafter for which no Doppler or any other quality filter is applied."

Furthermore, for practical application of the results the differences between uncorrected reflectivity (uZh) and Doppler corrected reflectivity (Zh) need to be shown and discussed. The guessing about the importance of the (mainly static) nacelle of the WT for the echoes is not necessary as a Doppler filter will separate echoes from the static parts of the WT from rotor echoes.

Actually we think it is a good idea to include some analyses for the Doppler filtered reflectivity. We do this for the 2020 campaign, where most of the other results are shown. However, we do not have spectral data (PSR) for the whole 2020 campaign to apply a filter in post-processing for the whole 22 days and every echo. PSR recording was limited by the signal processor. We tried to record it for 5 min, every 5 min to allow the data can be processed, stored and sent to our data archive. Most of the time it worked well, but for some periods it was still not working smoothly with the result of data gaps, occasionally some days.

With the new Figure 8b, we show the Doppler notch filtered (width +- 3.5m/s) reflectivity time series. To better see what orientations and rotor speeds are captured for both time series we included a new Figure 9, where we show in polar projection the normalized distributions of WT1 nacelle orientations in conjunction with the rotor speed.

Old Figure 9 (new Figure 10) is redesigned (also according to one of your other comments) now and shows the results only for ZH and filtered ZH for comparison. From the new scatter plot (Figure 11), it can be affirmed that our previous assumption of high returns from the nacelle was misleading and probably not correct in all situations. The differences are much less pronounced for the maximum than for the median or minimum. So to say the applied Doppler notch filter is more effective for the less high returns coming from the static parts.

We believe it is a nice addition and improvement of the manuscript preparation when the filtered ZH is included.

Page 11-13, Section 3.1 and 3.2: Added results and discussion of new filtered ZH data. Description about the filter which has been applied with the PSR data. Inclusion of new Figures: 8b, 9, 10b, 11 with necessary explanations.

The first part of chapter 3 introduces some theoretical terms, especially radar cross section (RCS). It starts from the standard discussion of an single, isolated target. Equation (1) is valid for an particle on the radar axis. If its position deviates from the main axis, the directivity pattern of the radar antenna has to be taken into account.

A WT is not small compared to the 3dB beam width. Here the limitations start. The RCS as it is derived here _is_ sensitive to the directivity pattern of the antenna. A different pattern would produce a different weighting of individual parts of the WT, resulting in a different RCS. Additionally, the WT is not small compared to the radial resolution of the measurements (75 m). Already a flagpole (i.e. a point in a PPI) contributes to two consecutive range gates if it is not perfectly positioned at the center of one range gate. As the diameter of the antenna is nearly twice the length of a range gate, the echo from the WT is distributed in three or four range gates if the relative yaw angle is 90 degrees or close to it. One range gate never contains all the echo from the WT. This needs to be explained and taken into account when deriving RCS from reflectivity.

In the following text and the figures the authors provide sometimes reflectivities, sometimes RCS values. As stated above, I do not trust the RCS and I do not want to calculate back an forth to compare RCS and Z values. My recommendation is:
(i) improve the derivation of RCS by discussing the effect of the spatial extend of the WT,
(ii) determine the offset between RCS and Z for each of the WTs, and then
(iii) stay to the measured reflectivities without introducing additional uncertainties from the conversion to RCS.

In general we agree with the reviewer comment here. Talking about RCSs is misleading. However we try to satisfy two different views on the subject: In the radar world reflectivity is not such a known concept as RCS is. To keep both communities (weather radar and radar) interested in the paper we keep to some extent the discussion about RCS.

We also agree that one range gate cannot contain the entire WT, but it does contain almost all the energy. The pole and the nacelle of the WT are the brightest scatterers and only the tips of the rotor would scatter out of a gate.

Regarding (ii), we introduce a new Table 5, where we present all 3 conversion factors F for each WT.

Regarding (iii), we still present the general conversion strategy/equations and minimize the discussion about RCS. It is not completely removed because people from the aviation radar world are interested in it as it was derived.

Regarding (i), we have to say that for the current study/paper it is out of the scope for us at the moment.

I do not follow the argument of the orientation of the nacelle to be important for the backscatter (line 288 and later). (i) The (scattering) area of the nacelle is small compared to the area of the rotor. (ii) The nacelle often has more or less straight "walls", leading to a stealth behavior: as long as the radar is not perpendicular to the nacelle surface, the (strong) echo is not scattered back to the radar. (iii) The scatter properties of a nacelle might be severely dependent on corners on the surface of the nacelle, which can have the effect of a corner reflector. Thus the properties of this individual nacelle are not representative for other type of WTs.

In order to prove the theory of strong nacelle echoes, the authors should discuss the effect of a Doppler filter. Such a filter should be able to suppress the nacelle echo, as the nacelle is (nearly) not moving.

Yes, we agree that our previous argument about the nacelle effects on the high returns were not fully thoughtful in all details. We thank the Referee for this important comment. We will remove the corresponding text passages, where we argue wrong, and include the more meaningful discussion about the Doppler filter effect on the echoes. In this we way, it can be more certain that the rotors of the wind turbine play the key role.

See also our comments to the second comment of Referee #1 here, where we discuss the Doppler notch filter.

Section 3.2: Additionally to comment 2, we follow the argumentation of the Referee about the influence of the nacelle in the text and note: "Regarding the possible influence of the WT nacelle, it can be argued that the effective scattering area of the WT nacelle is small compared to the area of the rotor. In the end this could lead to a stealth behavior, as long as the radar angle of attack is not perpendicular to the nacelle surface which is given for our measurement setup."
* * *
The last concern is about the linear correlation coefficients used in Fig. 11 and the text on it. In case there is a relation between WT physics and radar echoes it is not necessarily linear but probably strong. Assuming a linear relation between pitch angle and echo strength means - just to clarify my argument - a significant stronger echo at 390 degrees pitch angle than at 30 degrees. The surfaces on the rotor blade are in no way changing in a linear way.

We are facing of course a very complex problem of two independent movements of the WT slow yawing (turning the nacelle), which also changes the angle of attack towards the blades, and the blade pitch adjusting itself. If the WT nacelle relative position is at 0° or 180°, we expect for a blade pitch angle of for instance 90° higher returns than for a blade pitch at e.g. 30°. You are right the surface of the blades (which we do not know exactly) will probably not lead to a linear response. But in a first a approach, we were trying a simple linear correlation analysis.

We suggest to be more careful in the text, where it should be mentioned that the blade surfaces will likely not give a linear response. Unfortunately we do not have the expertise to model the EM interaction of WT blades and go into more detail.

Section 3.2: "A theoretically strong downhill linear relationship with r = -0.67 can be calculated for them, but it has to be taken into account that the complex blade surfaces will likely not give a linear radar echo response which in turn would initiate some misleading result."

With this argument, taking |sin| of the yaw angle is useful step. The images in Fig 12 bring some evidence that linear relations are not totally misleading. Nevertheless for |sin(yaw angle)| there is a significant shift to stronger echoes, not described by the linear relation.

Again here we suggest to include in the relevant text passage a note where we follow your argument that we see a significant shift to stronger echoes, not described by the linear relation.

Section 3.2: We add sentence: "It has to be added that a significant shift to stronger echoes is not described by the linear relation."

Fig 11 shows 28 correlation coefficients for three different samples of the measurements. Only 16 of them relate a WT related value (psi, theta, rs, and u) with a radar variable (Zmax, Zmin, delta Z, Zmedium). Wouldn't it be possible to show more of these scatter plots but only the four relations for two different samples?

Now we have already quite a lot of figures and even one more scatter plot for the comparison of the filtered ZH and thus we would in a first revision attempt not include more Figures. Please consider also the fact that we will try to publish more on this topic in the future where new  analyses will be presented. However Figure 11 is revised, by only showing now two samples only (rs >1 and rs<=1).

See change of old Figure 11 to new Figure 13 and inclusion of Scatter plot in new Figure 11.

In Figs 12 and 13 there are 95 % confidence intervals marked by shaded areas. But in no case 95% of the measurements fall into these intervals. What is the meaning of these 95% confidence intervals?

The regression lines and corresponding confidence intervals were produced with the *seaborn* Python package. For some parts, including  your point for the confidence intervals, the sub-function *regplot* has been used. Maybe there is a small misunderstanding here.  A parameter CI is passed to the function (in our case 95%) which defines the size of the confidence interval for the regression estimate (not the measurements). This will be drawn using translucent bands around the regression line. According to the *regplot* user guide, the confidence interval is estimated using a bootstrap method.

If we did not stated it clearly, a more precise definition of the meaning of these shaded intervals will be added to the captions of the new Figures 14 and 15. In the text in Section

3.2, we say now: "In addition, the linear regression lines with  confidence interval sizes of 95% for the regression estimate are drawn using translucent bands around the regression line"
* * *
Lately, the values for Zhmax are limited by saturation of the receiver as the authors mention. What does this mean for correlation coefficients of Zhmax to some other value?

If our measurements are facing a saturation issue (where we are not 100% sure of) it will of course affect the correlation coefficients. In the summary/conclusions part we say already: "Indeed, the observed discrepancy between the correlations of ZH,min / ZH,max and the blade pitch angle , for which ZH,min is much higher downhill correlated then ZH,max is uphill correlated, could be attributed to a receiver saturation issue." So if there is some linear relation to ZH,max, a possible saturation will reduce the correlation coefficients.

If for it is important to mention this issue earlier in the text, we suggest to include a short sentence in Section 3.2: "The possibility that ZH,max is facing some radar saturation issue during the 2020 measurement campaign, it could be the case that the correlation coefficients are lower than they should be."
* * *
Minor remarks:

L 23: 205 GW is the _installed_ capacity of WTs. As the wind is unsteadily blowing, the amount of yearly produced energy is significantly less than 205 GW times 8760 hours (=1.8 PWh). The real production was "only" 417 TWh. These 417 TWh correspond to 15 % of the electricity consumption of Europe. Do not mix up these terms. BTW: There is no proof that the number of wind turbines needs to increase. It might be the size o the turbines instead.

Thanks a lot for this hint, we add to the introduction the real amount of production. Regarding the number of WT in future, I would say that it is still likely that the number has to increase. But of course it is not proof-able. Typically I would argue that bigger turbines are more difficult to install and maintain. We are assuming a perfectly rational market here and green energy is becoming trendy so we expect more wind turbines to pop up, needed or not.

Our latest reference report is the following:
Nghiem, Aloys, Iván Pineda, and P. Tardieu. "Wind energy in Europe: Scenarios for 2030."
*Wind Europe*(2017): 32.

If OK for you, we change our text passage to and change to the latest reference mentioned in black above:

"The total capacity of wind energy in the EU in the end of 2019 is 205 GW. The effective real production amounts to about 417 TWh,  which is 15 % of the total consumed electricity. With

the fact that green energy is becoming trendy, a realistic outlook until 2030 is to have around 300 GW of wind turbine power installed (Nghiem et al., 2017). By assuming a perfectly rational market here, it is expected that more wind turbines pop up."
* * *
L 93: The directions towards the WT from the radar are 337.8°, 343.3°, and 340.2°. The RHI scans (according to Tab. 3) are only performed until 342.9° in fine resolution. Why does the range not cover the turbine at 343.3°?

The fact, that the fine resolution does not cover WT3 at 343.3° was simple a technical restriction of the radar control software, which does not allow more than a certain threshold of RHI numbers in one sequence: 30. So in the end it was not possible to cover all 3 WT in high resolution (0.1°) mode and a single volume. An additional constraint was the total scan time that we tried to minimize.
* * *
In Fig. 4d the positions of the most intense echoes are at roughly 338.5°, 341.5° and 343.3° which does not fit to the (first two) given WT locations. Same discrepancy for Fig. 5 and Tab. 4 also gives a different location of WT 1.

Note the following considerations:
What shown in old Fig.4 was for the year 2019, while the pointing scan is in 2020. Even if the site is exactly the same, we expect some minor misalignments.

At the beginning of the 2020 campaign we were running a couple of days with the PPI and/or RHI sequence like in 2019 and, based on this short dataset, we pointed to the location where the maximum return was observed. So it could be that it is not identical to the 2019 data (see also previous point).

Maybe it is also difficult in the Figure to identify the exact azimuth direction. Old Figure 4 is completely revised and split into 2 Figures 4 and 5  now. They will appear bigger and can be interpreted better in this way.
* * *
L 149: Sorry, I do not "clearly" see low ZDR and high RhoHV values at the wind turbine positions. Could you add an isoline of reflectivity at 35 dBZ to make it easier to see the effect? ZDR and RhoHV show an intense variation (noise) making it hard to assign individual pixels to a WT effect.

Indeed, in the PPIs the WT positions are difficult to see and we very welcome the hint to include the 35dBZ contour in the PPI plots.

In Figure 4, the isoline of 35 dBZ has been added in black color. Additionally we present the Doppler spectrum width W for the same scan. The corresponding text in Section 2.3 has been updated: "ZDR attains mostly slight negative numbers within the 35 dBZ contours. In

those areas and in the shadow behind, RhoHV is reduced to approximately 0.7 compared to the hill forest clutter in front of the wind turbines, where RhoHV reaches more than 0.95. For this specific example and the slow PPI scan, the Doppler spectrum width remains close to 0 m/s on average as indicated by the violet areas.
* * *
L 177: The inline equation contains two different Pr. One has the unit mW, one the nonunit dBm. The equation enables us to determine Pr. I know the problem to introduce dB and values on a logarithmic scale, but this way is wrong.

Totally right, if Pr has already been for linear unit in mW, we cannot use it again after Log-transformation.  We change to lower case for linear unit, and use lower case received power pr and pt (transmitted power) now.
* * *
L 179: sigma is not described as a scalar but deeply discussed as a function of the two angles occurring. What is a scalar is sigma(pi), the _back_scatter cross section.

We now use σb everywhere in the text and equations, because we are considering a monostatic radar, hence an angle of scattering equal to -pi (-180˚). In Equation 1, the backscattering cross section is indicated as σb  (hence, assuming a scattering angle of -180° for the angular RCS). Usually σb is expressed as a scalar and the dependence on wavelength is implicitly assumed: in our case $\lambda$ is 0.032m.

Section 3: Text passages are adjusted accordingly to the comment above.
* * *
L 180: I added a linefeed before "Precipitation".

We added a linefeed before "Precipitation".
* * *
L 192: In equation 4 the gain G0 looses its 0 without further notice. Why?

Just a mistake, thank you.

We correct and write in equation (4) G0 instead of G.

L 198: if you change equation 5 to sigma_i = pi^5 |K|^2 D_i^6/lambda^4, there is no need to change from sigma_d to sigma_i.

Yes, we will remove sigma_d and insert sigma_i instead. And update the text. As we always refer to a monostatic radar, we use now the subscript small b (-180° backscattering) in conjunction with sigma.

Section 3: "By substituting Equation 5 into Equation 4 the radar equation for spherical drops can be written as: ..."
* * *
L 201: You did not introduce Z, so you cannot convert from Z to RCS.

Right, we will correct this. When introducing Z (log transformed), respectivley z (in linear units) we will refer to Bringi and Chandrasekkar (2001):

#######
Bringi, V. N. and Chandrasekar, V.: Polarimetric Doppler Weather Radar: Principles and Applications, Cambridge University Press, https://doi.org/10.1017/CBO9780511541094, 2001
#######

Further we corrected an error in Equation 6, where we forgot the particle number per cubic meter n. In Equation 7 we now use lower capital z .
* * *
L 204: You should give all three conversion factors for the three WT.

Totally in agreement. We will give all 3 conversion factors in the revised version.

New Table 5 lists all the conversion factors for the 3 wind turbines.
* * *
L 248: "Averaged value"? How did you average? In dB or in m^2?

Old Figure 7 has been removed, it was not possible to reproduce it properly. This is our fault. Sorry. All the text corresponding to this Figure is removed, and your comment deprecated. In general, the part dealing with RCS will be shortened in the revised version.
* * *
L 248: The difference between two logarithmic values is always dB, not dBsm.

See comment above. Fig. 7 is removed.

L 280: Yes. The yaw angle of the WT is independent of reflectivity values. It is even independent of the presence of the radar. The message should be, the maximum reflectivities are insensitive to the yaw angle. - BTW: 10 dB difference is not really insensitive. It might be less intense than expected, but 10 dB is a factor 10.

True, our formulation was the wrong way around. It will be fixed. We include also your note, that the differences are less than 10 dB. At this point we do not define what is insensitive or sensitive. We give just the result, and our opinion that it is insensitive.

Section 3.2: "From the polar plots in Fig. 10a, we see that the maximum measured reflectivities are insensitive to the relative yaw angle of the nacelle. This is evident by the round and not too much disturbed distributions (less than 10 dB) of the color-mapped scatter points."
* * *
L 285: The variation of the maxima (in a) is in the order of 10 dB. The variation of the P99 values is also 10 dB. Why do you say, a directivity is more visible in b than in a? Same question occurs for the median values. - Plot (just for you) the figures into a Cartesian coordinate system and see if the messages is stable.

Here we mean with "first deformations become visible" the pattern at 90° and 270°, which are not as pronounced for ZH,max as for P99. In our point of view it gets clarified enough in lines 286-288.
* * *
L 417: See remark on line 248.

Thanks, the unit will be corrected to [dB] instead of [dBsm].

Figures:

The figures are a main drawback of this paper. It is the duty of the authors to provide figures the show - not hide - the information. Along all the figures only a few axis descriptions, titles and so on are readable. I do not want to use a magnifying glass to read the figures! Hardly any of these figures can be accepted as it is.

Additionally, figure 12 makes use of bright and dark green and blue. It might be that the printer of the authors (or their screen) shows these differences. Mine not. There are colors which are better distinguishable than bright and dark blue. There are different marks but + signs.

We apologize for the figures in the first manuscript. We tried our "best" and revised every single plot for a better readability. We do not use plus signs any more, except for the WT locations in Figure 1. As your wish, the colors in Fig. 12 changed and do not use light and dark color modes anymore.

As a hint, I think the Figures in the final publication version (if accepted) will always appear bigger, because of the constraints of the manuscript mode of *Copernicus*. They advise the Figure width to not use the whole page width in 2 column mode (e.g. compare width of caption text).

Fig 1. The black + for the radar position is totally invisible in my printout in subfigure a. Same for the border lines in both figures. Axis labels and tic labels are fine for a and b, tic marks for c are at the limit.

The radar sign is now red, while the WT signs are white. Also the colorbar changed to Python native (*viridis*). Border lines are now better visible. For the 2 top plots, the all font sizes were increased. The cross-section is produced by SWISSTOPO website and we cannot improve it a lot. I tried to increase the whole subplot a bit.
* * *
Fig 2. Absolutely unreadable in printout. On the screen at 400 % size acceptable but the velocities at the color codes.

All fonts in Fig. 2 adjusted, color bar changed and the bin width was increased to 4 m/s. It is much better readable now.

Fig 3: Axes labels and tic labels at the limit. Sketch of wind turbines is a good try to explain nacelle orientations. I think, a view from the top is easier to see.

Axes and tic labels are enlarged now. For a good reason we did not touch the WT sketch of Fig.3. Were are no CAD experts and it would simply cost too much time for us to adjust (for in our opinion little gain).
* * *
There is no additional information but only more confusion in showing absolute and relative orientations of the WT. Simply give the offset angle and show one of them.

Here, we totally agree with the Referee comment, that it is not necessary to show both, absolute and relative orientations. We stick now to the relative ones and provide the offset angle, which is 158.9°. In addition all font sizes increased for Fig. 3b and c.
* * *
What is the source of the echoes left of WT 1 in d and e?

I have to assume you mean Figure 4d and e. Old Figure 4 is divided now into Fig. 4 (PPI) and Fig. 5 (range span plots). The strong echo left of WT1 has most likely its origin in ground-clutter backscatter as the given WT surroundings consist of a hill forest. These echoes are coming from a shorter distance than WT1 (see range span plots of $ZH,max$). Now an additional range span plot for the PPI scans is included. In total Fig. 5 consists now of 4 subplots.

In Section 2.3 we write accordingly: "With the distance of these range gates, it is obvious that the high returns left of WT1 are not associated to a turbine effect as they appear at a shorter distance from the radar of about 7500 m compared to the WT1 distance of 7740 m. Given the surroundings consisting of uphill terrain (s. Fig 1), the high radar returns have most likely the origin in background clutter from the hill forests."
* * *
Fig 4: Can you provide a PPI of spectral width as well? Labels in a through c at the limit, in d through f totally ridiculous.

Fig. 4 is completely revised. See also our last comment. In the new Fig. 4 we also show now the Doppler spectrum width. All labels better visible now.
* * *
Fig 5 and 6: If you assess the patterns around the wind turbine to the directivity pattern of the radar antenna, we need that directivity pattern to be shown in the same form. Is there any difference between the top rows and the bottom rows but an offset? If not, bottom rows are not needed.

We see, that the RCS plots in those Figures are not strictly needed, as there is just an offset to the ZH,max. This offset is now given in Table 5 for all 3 wind turbines. The RCS plots here are removed. The new Figures 6 and 7 show only the statistics of ZH. Currently we are not able to provide a directivity pattern of the antenna in the same form.
* * *
Axis and tip labels are below the limit.

Axes and tic labels of Fig. 6 and 7 are bigger now.
* * *
Fig 7: In a, c and f the local maximum at the position of the WT is a bit larger then the overall maximum around the WT. This is not possible by definition.
Axis, tip and legend labels are totally unreadable. Digitally they appear to be readable at roughly 250 %.

Figure 7 is discarded in the manuscript.
* * *
Fig 7/8: One figure shows RCS in dBsm, one figure shows reflectivity in dBZ. This is not helpful. Please show always Z in dBZ.

In general we follow the suggestion of the Referee and stick more to Z in all plots. Nevertheless RCS will be mentioned still in the text parts, where appropriate.

Fig 8: Axis and tip labels at the very limit.
Moving average in dB, not in mmˆ6/mˆ3?!

In Figure 8 we show now both the filtered and unfiltered data time series, as discussed earlier in the major remarks of Referee #1. All fonts enlarged. The red scatter points show the moving median and it should be still in unit dBZ. We do not average here.
* * *
Fig 9: Labels and so on much too small.

The figures are strongly misleading! Whereas the reflectivity scales show a range of 60 dB, the cross section scales only show 40 dB. This changes the shape massively!. Max reflectivity at 110∘ is roughly 80 dBZ, at 140∘ it is 70 dBZ, 10 dB less. Max RCS at 110∘ is fairly 45 dBsm, at 140∘ it is 35 dBsm, also 10 dB less but the "bulge" looks much stronger - it is not.

The median reflectivity has a difference of 68 dBZ (170∘) down to 56 dBZ (310∘), i.e. roughly 12 dB. The median RCS varies between 32 dBsm (170∘) and 20 dBsm (310∘), also 12 dB. All the difference that are visible between a/b and c/d are caused by the way of presentation.

I recommend to combine a and b into one figure (having three dot circles instead of two, showing only values between 80 dBZ (max value of all) and 50 dBZ (min value of median values).

Old Figure 9 is redesigned into Fig. 10 to mostly follow this comment. The RCS version is completely omitted. The P99 is included as a thirs color map circle. For comparison we now show the version for unfiltered ZH and Doppler notch filtered ZH. Both plots have the same y-axis limits between 30 and 80 dBZ. We had to go down to 30 dBZ because of the lower filtered ZH values.

New discussion and adjusted text passages in Section 3.2.
* * *
Bin width are differences and thus always given in dB,
not in dBZ or dBsm (see comment on line 240).

You are right, we correct this error.
* * *
Fig 10: "WT1 blade angles" means pitch angles?

Yes, with blade angles we mean the pitch and not yawing of the nacelle.

Fig 11: Do we need more than the lower 4x4 CC values? Probably instead of simple linear correlation coefficients scatter plots could tell more.

We suggest back to only show 2 samples instead of 3, but we stick to the heat-map structure, as it is given out nicely by *seaborn* already and it is not very well adjustable unfortunately. All labels are better visible in the new version. The point of showing more scatter points will be postponed to a future study where also polarimetric variables will be a point of interest. In addition we think we are already at the limit of number of figures in this paper.
* * *
Fig 12: Especially the legend is unreadable. Bright and dark green, bright and dark green are indistinguishable in my printout. Choose other colors.

Sure, we thank you for this note. Everything should also be distinguishable in a printout. The former light colors in Fig.12 (new Fig. 14) changed to dark yellow and red. In our printouts it is satisfactory in this way.
* * *
Fig 14: Legend unreadable.

Legend font size is larger now.
* * *
[Figure]

[revised manuscript text omitted]

---

## Author Comment (AC2) · 12 Mar 2021

Martin Lainer (on behalf of all co-authors), 12.03.2021

Color Code: **Referee comments**, **Author responses**, **Relevant changes in the manuscript**

We thank the Referee for his constructive and well structured comments.

In the following part we will give point by point answers and/or comments to all points raised by the Referee. We try our best to further improve the paper based on the review. Relevant changes in the manuscript will be explained/cited with highlighted green color coded text. For simplicity we only provide links towards the section of the manuscript. For the detailed changes please refer to the visual mark up version produced by *latexdiff*. Be aware that the visual mark up did not work for all the Figures and their captions smoothly as well as for the references.
* * *
My overall statement: I find this paper very valuable, since it develops in a comprehensive form the deviation of a single WT's RCS from a a measured dBZ gained from a precipitation radar. Special care is taken to set up a measurement to isolate the radar return from specific wind turbines. Much work is invested to present a meaningful statistical analysis.

Remark on nearly all Figures: The tics and axis names are to small, and some important marks in the diagrams are also too small. This makes some figures quite unreadable.

Many thanks for the good rating of our performed study. Regarding the point of the Figures, all of them are improved in the revised version, regarding e.g. the readability (labels, legends, tics, color codes).
* * *
Abstract (Paragraph 1) and Introduction (i.e. Par. 45):
The radar in use is a "precipitation radar". Please use at least once this term.

We were using the term weather radar mostly. Weather radar + X-band defines in our opinion pretty well the system and its capability. It could be mentioned once that this weather radar is sensitive mostly to hydrometeors in the precipitation-size range.

In Section 2 we now say: "The measurements presented in this paper have been collected with a dual-polarization, simultaneous transmission and reception (STAR), mobile Doppler weather radar, which operates at a frequency of $9.48$\,\unit{GHz} (X-band). This radar system is sensitive mostly to hydrometeors in the precipitation-size range.".

In the introduction we use once more the expression X-band in conjunction with the weather radar system.
* * *
Par. 30:
"RCS is an optimal variable to estimate the effect of a wind turbine on the performance of a radar system, in fact existing numerical models for estimating the back-scattering efficiency of wind turbines rely on this quantity."

According to [1] ("scattering cross-section:") the incident field is assumed to be planar over the extent of the target. This is in principle not the case for objects on the ground and causes some restrictions.

We think it is more appropriate to focus on the reflectivity measurements in the paper (as pointed out by Referee #1). Still we would like to present the RCS as an generalized concept which can be more of use and interpreted by the radar community rather than the weather radar community. We suggest to clarify within Par. 30:

"RCS is an optimal variable to estimate the effect of a "point target" on the performance of a radar system. With the term "point" we mean a target, which is much smaller than the radar sampling volume and with a size such that the incident field could be assumed to be planar over the whole extent of the target. However, the RCS concept is often generalized and extended to larger objects, starting with small airplanes, but then reaching even large airplanes and wind turbine. As a matter of fact, existing numerical models for estimating the back-scattering efficiency of wind turbines rely on this quantity. It is the projected area needed to isotropically re-irradiate the same power as the target scatters in the direction of the receiver and is usually expressed in decibel units related to one square meter (dBsm) (Knott et al., 2004; Skolnik, 1990). The detailed background on how the RCS is computed within our system is given in Sec. 3."
* * *
70:
I am missing a hint or a justification why the radar pulse width was selected 0.5µs but no other values. Is there a technical restriction? Same for PRF.

The overall requirements for us to be met were to have a maximum range resolution but still containing most of the WT object and a high as possible Nyquist velocity. With a pulse width of 0.5µs  we achieved a good target range resolution of 75m. Compared to the 0.33µs pulse shape, the one for 0.5µs is also more uniform and thus preferred in our system. It has been decided to keep this value constant for all the measurements. The antenna movement is slow, ensuring data collection every 0.1° (one radial) azimuth, while the PRF is set high enough to ensure a large number of pulses for each radial and a reasonably good unambiguous velocity (Nyquist) range.

Section 2: we add correspondingly the following: "For the campaigns we are consistent and stick to one pulse width of 0.5µs to have a good target range resolution of 75m, where most

of the radiation energy is scattered by the WT object. The 0.5μs pulse shape is, compared to the one for 0.33μs, more uniform and thus preferred in our system. The antenna movement for the measurements in 2019 is slow, ensuring data collection every 0.1° in azimuth (one radial), while the PRF is set high enough (2kHz) to ensure a large number of pulses for each radial and a reasonably good unambiguous velocity range."
* * *
125:
"... co-polar correlation coefficient are represented on 2019-03-24."
Are these coeffecients in use later when developing eq. (7) using eq. (3) ? (3) is defined for circular polarization.
"During the fixed-pointing scans, the antenna of the radar was not moving and always pointing to the same wind turbine"

Yes, to derive equation 7, we have used the Probert-Jones approximation of the sampling volume.
* * *
Fig. 2 too small, unreadable

Figure 2 is redesigned. A different wind speed bin width of 4 m/s is applied  and the color bar changed. Now the legend is much better visible.
* * *
150:
"The wind turbine clutter is clearly visible at the range from 7.7 to 8.6km as high ZH, HV and low ZDR and all three turbines can be distinguished from other ground clutter signals"

In order to make the WT clutter better visible, we plotted (regarding also the comment of Referee #1) the 35 dBZ contours on all PPIs. It is the fact, that with ZDR it is very difficult to distinguish the WT from the surrounding clutter echoes. The expressions in the text on it, are now more carefully formulated.
* * *
165:
"In the following part of the paper, all the horizontal reflectivity ZH measurements and retrieved RCS data ..."
See restrictions of RCS mentioned above and discuss the applicability.

We will delete the text part "...and retrieved RCS data ...". Also Referee #1 mentions the restrictions of RCS for such a huge target. Hence, we would keep a low profile and focus on the measured radar reflectivity values and simply write the conversion factor for going from dBZ to dBm2 , which is , for instance, ~34.4 for WT1 at 7740 m distance.

Section 3: Added Table 5 with all 3 conversion factors to go from dBZ to dBm2. Section 3.1: removed some parts where we explicitly mention RCS.
* * *
175:
eq. (1) is simply a form of the radar equation and should be referred to as it is.

We suggest to write something like:

Section 3: "According to Battan (1973) the radar equation for a single, isolated target is: ... where [Pr] = mW is the received lossless power by a directional antenna."
* * *
185:
"Accounting for the actual distribution of power within the beam generated by a circular parabolic antenna, a correction factor of 1/(2 ln2) was introduced by Probert-Jones (1962):"
The development of the later eq.(7) has some uncertainties, i.e. the usage of eq. (3) which applies to circular polarization. Please mention that and discuss what the restrictions may be applying linear polarization.

The main focus of this paper is on the backscattered radiation received by the channel that is sensitive to the horizontal polarization. Your question regarding what state of polarization is actually transmitted by our X-band radar is intriguing and interesting; however, such discussion goes far beyond the scope of the present analysis and study.

What we can tell you, is that we are far from being sure that the relative phases of the simultaneously transmitted H and V components are perfectly adjusted (0˚ shift) transmit purely linear slant polarization (45˚ or 135˚). On the contrary, some practical measurements, which have been performed at the Site Acceptance Test with a spectrum analyzer and a receiving horn which was rotated by 45˚ steps (horizontal, vertical, and two slant-observations ), seem to show that the X-band radar is transmitting something that is more similar to circular than to linear slant.
* * *
195:
"The Rayleigh approximation of the backscattering cross section of a single water drop d can be expressed as ..." Is there a reference to literature?

An appropriate reference to literature would be the book by e.g. Fabry, F. Radar Meteorology: Principles and Practice, 1$^{st}$ ed., Cambridge University Press: Cambridge, UK, 2015. In particular, page 11-13, Sec. 2.2.2 Scattering Regimes and 2.2.2.1 Objects with sharp boundaries.

Another recent good book is by Ryzhkov and Zrnic: Radar Polarimetry for Weather Observations, Springer, 2019.

We add the citations mentioned above for the equation (5).
* * *
215:
"The statistical overviews in Figs. 5 and 6 show the maximum (1st row), median (2nd row) of ZH and the maximum retrieved RCS (3rd row) for all three wind turbines."
So equation (7) is used for that representation, please make a note.

You were right, Eq.7 was used to derive the RCS for the Figures 5 and 6. However in regard to the changes ma
* * *
250:
"Figure 7 shows the variability of the location and intensity of the RCS maxima. The farther apart the blue and black histograms are, the higher is the variability of the distribution of the maxima."

Please explain this in more detail.

In Figure 7, both curves represent RCS values according x-axis. However, the legend says "Pmax" and "RCSmax", which is confusing. Please correct or clarify.

"The blue histograms are calculated at a fixed position (as listed in Table 13)"

Mistake: Table 13 ?? –> Table 3

Unfortunately Figure 7 had to be removed from the manuscript and thus all comments on it are omitted. The reason is already explained in the answer to Referee #1.
* * *
265:
From the 2020 measurement, it may be interested to zoom much deeper into time to see the fluctuation across a short period, i.e. within some seconds in contrast to the long-term diagram as of Fig. 8.

Figure 9: "dBZ" and "dBsm" refer to absolute values. If the meaning is a variation, i.e. for a bin, always use "dB" alone without reference!
Same mistake as in line 265 "about 4dBsm higher". There are maybe more of these in the entire text. In line 285 however, it's correct: "... bin width of 0.5dB".

Regarding your first comment here, we note that a future study will go into and analyze much shorter time periods, where very interesting effects could be observed also in context

to polarimetric parameters. In this manuscript, where we give a first overview about the campaigns, we would not go yet into such details. But totally true it is of high interest to the community.

We will go through all units, to be sure that variations/differences etc. will be shown correct as dB. Thanks for the careful review on this.

All units have been checked and corrected.
* * *
280:
"From the polar plots in Fig. 9(a, c) we see that the relative orientation of the turbine/nacelle is insensitive to the maximum values, evident by the round and not too much disturbed distributions"

Is this correct for a) _and_ c) ? For a) I agree. I think for the median values (red) this is correct for a) and b) depicting the dBZ. Is the averaging time always the 10 min as of Fig. 8 ? In sub-figures c) and d) the difference of medium RCS across the two positions "radial 90/270°" and "tangential 0/180°" is lower than I would expect.

Could that be validated from other sources/measurements for X-Band (references)? I have seen many RCS simulations for WTs in C band that claimed differences >20dB. From own measurements close to reflecting WTs we got differences of >30dB with rotor planes in frontal (0°) and side (i.e. 90°) view [2]. However, there is obviously a difference from the radar's point of view. It observes the target over a larger distance that includes terrain effects.

Old Fig. 9 (new Figure 10) changed regarding to Referee #1. There we now do not show RCS anymore, but a version with Doppler filtered data to separate the static WT clutter parts. Between (a) and (c) in the old Figure only an offset regarding added Table 5 is present. The time window for the median values in Figure 8 was 10 min. In the polar plots we use all available observation as shown in black/grey in new Fig. 8a and b. Some uncertainty is introduced because we have the orientation only every 10 min. For a direction of 90° for instance we assign all ZH measurements to and then calculate the median from the whole sub data set.

To find an appropriate source for validation purpose is difficult, as the measurement setup (point of view, distance, clutter) is quite unique. Accommodating also with your last comment, we suggest that we cite your study [2] on the C-Band results.

290:

"Future work should take also into account numerical simulations ..." Yes, that might be interesting. I have not seen many in the X band. However, methods like MoM or even MLFMM might not be suitable due to the huge number of triangles to work with. Asymptotic methods l(like PO, GO), however, might neglect some eletromagnetic (near) field effetcs.

Thank you for this short overview of available methods that could be applicable. Colleagues from the University of Bern perform MoM calculations with HFSS. In future we might consider a small collaboration on the WT subject.

300:

"For example on 2020-03-12 or 2020-03-22 blade pitch angles of WT1 were aligned at roughly 90° over hours..."

In Fig. 10 it would be helpful also to have an additional curve that simultaneously shows the orientation across the time axis.

You are absolutely right, it is a very good idea too show in old Fig. 10 (now Fig. 12) the time series of the WT1 relative orientation. In order to keep a good overview we decided to not plot an additional curve into Fig 12a. Instead we show the exact same time interval with aligned tic labels in subplot 12b underneath.

As written, we included an additional subplot in Fig 12, that shows the relative nacelle orientation.
* * *
345:

Why is the relative yaw angle Psi normalized, and the other angles are not?

For PSI it made sense from a geometrical point of view  as if any linear response would be present, it would be destroyed by the left/right and forward/backward symmetry (polar view). For the blade angle it was not necessary as they only go from about 0 to 90°. Still it is questionable (and we raise this point in the paper now) if any linear is present given the complex structure of the blade surface. The blade surface non-linearity is shortly discussed in the revised manuscript (s. also comments by Referee #1).
* * *
410:

Ground radars in X (or K) band are used in aviation only as SMR (surface movement radars) or for the pecision approach radar (PAR). Unfortunately, for surveillance or air defense radars in S band, the actual antenna beam widths are much too large to discriminate single WTs at a far distance. Those measurement results, obtained from a similiar measurement campaign, would be really interesting.

We are fine with your comment, and suggest to mention the following here:

Section 4: "The maximum returns are describing a general worst case scenario, which is of interest, for example, to the mostly civilian aviation safety sector when measurements from K- or X-band SMR (surface movement radars) or PAR (precision approach radars) are validated. For the military aviation sector it would also concern radar systems for guidance and surveillance, both for ground-based and airborne systems."
* * *
425:
"... the computational requirements are huge..."

Yes, and be aware of the electromagnetic solvers methods and their limitations (mentioned above).

Yes, for simple user like us, it is not easy to thoroughly understand the limitations behind the necessary assumptions of the various numerical techniques when solving Maxwell equations. For now we try not to go into more detail regarding possible simulation techniques in the current paper.
* * *
430:
"Thus dedicated measurement campaigns with e.g. mobile radars offer another approach to assess wind turbine reflectivity and RCS in a broad range of real environmental scenarios"

Yes, this is necessary. As an example, We did that for C band precipitation radar [2].

As written before we would like to cite your interesting study [2] with the C-Band measurements here.

Section 4: We include: "Bredemeyer (2019) used an UAS (unmanned aerial system) as a passive bistatic radar (PBR) at nacelle altitude for reflectivity measurements of a wind turbine in the C-band (5.64 GHz). From short distances below 300 m they got RCS differences of more than 30 dB between WT rotor planes perpendicular and parallel to the PBR."
* * *
########################################################################

[1] IEEE Std 211-1997: IEEE Standard Definitions of Terms for Radio Wave Propagation. https://ieeexplore.ieee.org/servlet/opac?punumber=5697

[2] J. Bredemeyer, K. Schubert et.al.: "Comparison of principles for measuring the reflectivity values from wind turbines", International Radar Symposium - IRS 2019, Ulm, Germany